# The CBL1/9-CIPK1 calcium sensor negatively regulates drought stress by phosphorylating the PYLs ABA receptor

Zhang You[1], Shiyuan Guo[1], Qiao Li[1], Yanjun Fang[1], Panpan Huang[1], Chuanfeng Ju[1] & Cun Wang ®[1,2] ✉

The stress hormone, Abscisic acid (ABA), is crucial for plants to respond to changes in their environment. It triggers changes in cytoplasmic $Ca^{2+}$ levels, which activate plant responses to external stresses. However, how $Ca^{2+}$ sensing and signaling feeds back into ABA signaling is not well understood. Here we reveal a calcium sensing module that negatively regulates drought stress via modulating ABA receptor PYLs. Mutants *cbl1/9* and *cipk1* exhibit hypersensitivity to ABA and drought resilience. Furthermore, CIPK1 is shown to interact with and phosphorylate 7 of 14 ABA receptors at the evolutionarily conserved site corresponding to PYL4 Ser129, thereby suppressing their activities and promoting PP2C activities under normal conditions. Under drought stress, ABA impedes PYLs phosphorylation by CIPK1 to respond to ABA signaling and survive in unfavorable environment. These findings provide insights into a previously unknown negative regulatory mechanism of the ABA signaling pathway, which is mediated by CBL1/9-CIPK1-PYLs, resulting in plants that are more sensitive to drought stress. This discovery expands our knowledge about the interplay between $Ca^{2+}$ signaling and ABA signaling.

Global warming poses a severe threat to crop survival and yields, therefore, elucidation of the molecular mechanisms through which plants respond to drought is crucial for food security[1]. Abscisic acid (ABA) is a phytohormone that is vital in coordinating various responses of plants to the external environment, such as drought, salt, and heat resilience[2,3]. The positive and negative regulators of the ABA signaling pathway have been reported. In the core ABA pathway, the perception of ABA by pyrabactin resistance 1 (PYR1) and the PYR1-like (PYL)/regulatory component of ABA receptor (RCAR) proteins (hereafter referred to as PYLs) leads to the binding of ABA receptors to clade A protein phosphatases type-2C (PP2Cs), such as ABI1, ABI2, HAB1, HAB2, PP2CA, and AHG1[4–7]. These effects relieve the inhibition of sucrose non-fermenting 1-related subfamily 2 protein kinases (SnRK2s) or other protein kinases, thereby activating the ABA signaling pathway, such as phosphorylation of slow anion channel-associated 1 (SLAC1), fast-activating anion channel

1 (QUAC1), and transcription factors (ABA-associated factors (ABFs))[8–11]. ABA also influences seed germination as well as dormancy and plays a vital role in plant growth and development[2,12,13]. In stomatal movement, the key event downstream of ABA is the stimulation of reactive oxygen species (ROS) production, which induces $Ca^{2+}$ influx from the plasma membrane and activates the $Ca^{2+}$-dependent ABA signaling pathway[14–20].

The *Arabidopsis* PYLs family consists of 14 members that are highly redundant and play various roles in plant growth, development as well as responses to stress[6,7,21]. Among them, apart from PYL13, in the absence of PP2Cs, monomeric PYLs (PYL4-6 and PYL8-10) have a higher binding affinity for ABA than dimeric PYLs (PYR1 and PYL1-2). The monomeric PYLs can partially inhibit PP2Cs in the absence of ABA[22,23]. The precise mechanisms by which these ABA-independent ABA receptors maintain their inactive status in the absence of stress are poorly understood.

[1]National Key Laboratory of Crop Improvement for Stress Tolerance and Production, College of Life Sciences, Northwest A&F University, Yangling, Shaanxi 712100, People's Republic of China. [2]Institute of Future Agriculture, Northwest Agriculture & Forestry University, Yangling, Shaanxi 712100, China. ✉e-mail: cunwang@nwafu.edu.cn

Given the significance of PYLs in ABA perception in plants, post-transcriptional regulation of PYLs has been intensively studied. The TOR kinase phosphorylates conserved sites of PYLs to inactivate them and balance plant growth as well as stress responses[24]. The AEL casein kinases promote PYLs ubiquitination and degradation via phosphorylation[25]. Postdegradation sorting of PYLs into vacuoles is mediated by the canonical endosomal sorting complex required for transport (ESCRT) components, such as VPS23A, FYVE1, and ALIX[26–28]. The E3 ubiquitin ligase (XBAT35.2) regulates ABA signaling by ubiquitinating VPS23A degradation. This ubiquitination is finely regulated by deubiquitinating enzymes (UBP12 and UBP13)[29]. Moreover, Cytosolic ABA Receptor Kinases (CARKs) positively regulate ABA signaling by phosphorylating PYLs[30]. Various key protein kinases that have the ability to phosphorylate ABA receptors have been identified; however, their phosphorylation modification mechanisms have yet to be fully established.

Spatiotemporally distributed calcium ions ($Ca^{2+}$) are ubiquitous second messengers that play an important role in eukaryotic responses to biotic and abiotic stresses[31]. Physiologically, CNGC5/6/9/12 mediates ABA-induced influx of extracellular $Ca^{2+}$, increasing cytoplasmic $Ca^{2+}$ concentrations or ABA-specific oscillations[32]. These $Ca^{2+}$ signals are sensed by intracellular calcium sensors, such as CAM, CML, CDPK (also known as CPK), and CBL-CIPK[33–35]. The regulatory network for CBL-CIPK is crucial for plants to respond to adverse environmental stimuli[36]. Moreover, CBL-CIPK has been implicated in ABA and drought responses. *Arabidopsis* CIPK1, CIPK3, CIPK8, CIPK11, CIPK14, CIPK17, and CIPK23 are involved in ABA signaling[37–41]. The CBL9-CIPK3 complex negatively regulates ABA responses during germination[42]. Overexpression of *CIPK11* results in a drought hypersensitivity phenotype, while co-expressions of CBL5-CIPK11 in oocytes activate the SLAC1 anion channel[43,44]. Both *cbl1/9* and *cipk23* mutants exhibit drought resilience and ABA hypersensitivity outcomes in stomata, while CBL1/9-CIPK23 phosphorylates and activates SLAC1 and SLAH3 in oocytes[40,45]. Elucidation of the regulatory circuits involved in this paradoxical phenotype and the associated mechanisms will be of vital significance. These CIPKs have a role in ABA responses; however, their precise regulation processes have yet to be fully established.

In this study, we found that the calcium sensor (CBL1/9-activated CIPK1) phosphorylates PYLs by directly interacting with ABA receptors (PYR1/PYL1-6). The phosphorylation of PYLs inhibits their activities and negatively regulates ABA signaling. Additionally, Ser129 in PYL4 was shown to be essential for PYLs to bind ABA and inhibit PP2C. This negative regulation is important for preventing stress signaling by ABA-dependent and ABA-independent PYLs in the absence of stress. Under stress conditions, ABA inhibits PYLs phosphorylation by CIPK1, activating downstream ABA signaling and ensuring plant survival. The discovery of the regulatory mechanisms of CBL1/9-CIPK1 in the ABA signaling pathway sheds light on the intricate ways in which plants respond to drought stress and adapt to ever-changing environmental conditions.

## Results

In *Arabidopsis* cv. *Wassilewskija* plants, CIPK1 regulates ABA signaling[38], thus, we evaluated whether it responds to ABA signaling during drought in Columbia plants. We used two independent T-DNA lines, *cipk1-2* (N822018) and *cipk1-3* (GK230-D11), for our investigations. The expression pattern of CIPK1 was comparable to those of CBL1/9[38], and the expression of CIPK1 in guard cells was comparable to those of CBL1, CBL9, and CIPK23[40], which is consistent with our findings for native promoter-driven GUS transgenic lines (Fig. 1a).

Given the significance of CIPK1 in ABA signaling and its expression in stomata, we aim to assess its function in preventing water loss following soil drying. In this assay, 2-week-old seedlings were water deprived for 20 d, followed by rewatering. To ensure uniform drought exposure for all plants, the relative water content of the soil in each small pot was determined during the drought phase. No significant difference was observed in soil water content among the genotypes during the entire experiment (Supplementary Fig. 1a). Compared to the wild type, the two *cipk1* mutants exhibited better growth performance during late stage of soil drought, suggesting that the functional deficiency of *CIPK1* has greater drought resilience (Fig. 1b). To explore whether the improved water retention capacity in *cipk1* plants is linked to changes in transpiration rate, we monitored the rate of water loss from detached leaves. Our findings revealed that *cipk1-2* and *cipk1-3* exhibited a slower water loss compared to the wild type (Fig. 1c).

Further, the effects of ABA on stomatal closure were evaluated. The stomata were fully opened in MES buffer under strong light and treated with 10 µM ABA. The *cipk1-2* and *cipk1-3* stomata were shown to close faster than the wild type (Fig. 1d, e). We then utilized a far-red camera to measure leaf temperatures, which are indicative of transpiration levels, with higher temperatures, indicating lower levels of transpiration. Our results revealed that the *cipk1-2* and *cipk1-3* mutants displayed higher leaf temperatures than the wild type (Fig. 1f), suggesting that the *cipk1* mutants had lower levels of water transpiration compared to the wild type.

To validate the functions of CIPK1 in ABA-induced stomatal closure, a constitutively active (CA) variant of CIPK1 was generated under $Pro_{GC1}$ with the critical Thr (T) phosphorylation site in the activation loop converted to Asp (D)[46]. In this study, $Pro_{GC1}$: $CIPK1^{T179D}$ is referred to as *CIPK1CA*. Transgenic plants expressing *CIPK1CA* had lower water retention capacity after soil drying (Fig. 1g, and Supplementary Fig. 1b). We also conducted detached leaf water loss experiments and observed that the *CIPK1CA* plants had a higher rate of water dissipation compared to the wild type (Fig. 1h). Furthermore, our findings revealed that the ABA-promoted stomatal closure was impaired in *CIPK1CA* plants as compared to the wild type (Fig. 1i, j). In addition, the leaf temperature measurements indicated that the *CIPK1CA* lines had a lower temperature than the wild type, suggesting that the *CIPK1CA* plants transpired more water than the wild type (Fig. 1k). These findings imply that CIPK1 negatively regulates ABA responses during drought.

Physiologically, CIPK is activated by binding specific CBL, which initiates the downstream signaling processes[47,48]. Therefore, soil drought assays were performed on two CBLs (CBL1 and CBL9) that interact with CIPK1 in vivo and whose double mutants are sensitive to ABA, as well as the *cbl1/9/cipk1* triple mutant. The *cbl1/9/cipk1* triple mutants exhibited comparable water retention capacity to *cbl1/9*, consistent with previous studies[40], indicating that the CBL1/9-CIPK1 complex negatively regulates *Arabidopsis* responses to drought (Fig. 1l, and Supplementary Fig. 1c). Accordingly, the leaf temperature measurements also showed that *cbl1/9/cipk1* had similar water dissipation rates as *cbl1/9* and *cipk1* (Fig. 1m). In conclusion, the CBL1/9-CIPK1 complex negatively regulates ABA signaling during drought.

### CIPK1 interacts with the PYLs ABA receptors

The CBL1/9-CIPK1 complex has been shown to be a negative regulator of plant responses to drought resilience, and VaCIPK02 has been reported to directly or indirectly regulate the response to drought resilience by interacting with VaPYL9[49]; thus, we aimed to evaluate whether CIPK1 directly interacts with the ABA receptors, which are core components of ABA signaling. To validate our hypothesis, a bimolecular fluorescence complementation (BiFC) assay, in which CIPK1 harbors the N-terminal YFP (YFP^n-CIPK1) and the ABA receptors are fused to the C-terminal YFP (PYR1/PYLs-YFP^c) was conducted. When CIPK1 was combined with 14 PYR1/PYLs in *N. benthamiana* leaves for 4 d, the YFP fluorescence was detected in PYR1/PYL1-6 (Fig. 2a, and Supplementary Fig. 2a). To further investigate the location of interactions between CIPK1 and PYLs in plants, we conducted an analysis using tobacco leaf protoplasts. Our results showed that the interaction between CIPK1 and PYLs occurred in the cytoplasm, nucleus, and cell membrane (Fig. 2b). In addition, we also introduced CBL1-FLAG into

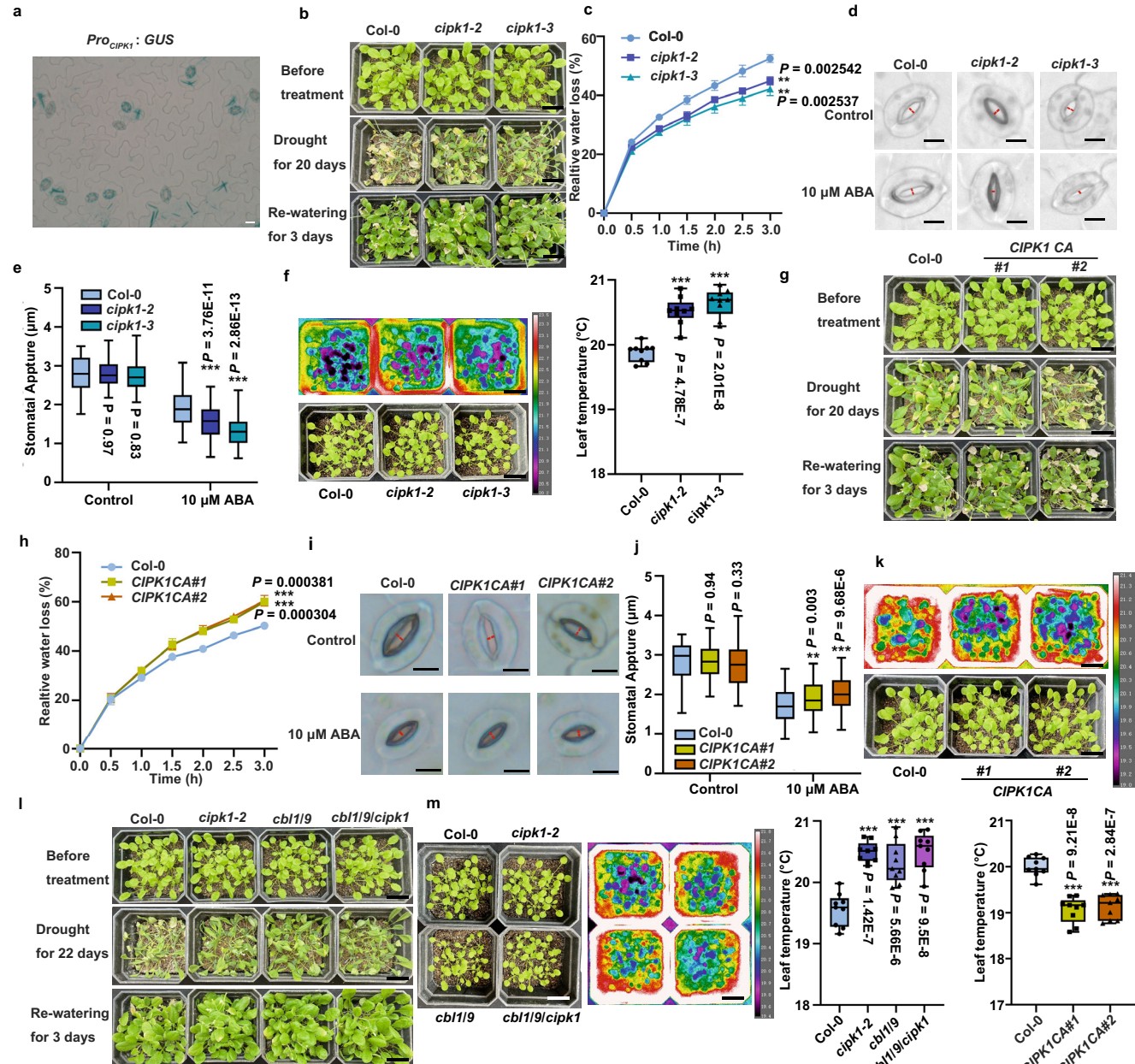

**Fig. 1 | Phenotype of *cbl1/9, cipk1* mutants, and *CIPK1* overexpression lines.**
**a** GUS Histochemical staining in *Pro_CIPK1* : *GUS* lines. The experiment was replicated thrice with similar results. **b** *cipk1* mutants showed enhanced drought resilience compared to the wild type. The experiment was replicated three times yielding similar results. **c** Water loss measurement in detached rosette leaves of *cipk1* plants. The experiment was repeated three times with independent treatments. Data are expressed as mean ± SD. **d** ABA-mediated stomatal closure in *cipk1* mutants. Stomatal width measurements are marked in the figure using red dashed lines. **e** Statistical analysis of ABA-induced stomatal closure in wild-type, *cipk1-2*, and *cipk1-3* plants. The experiment was repeated three times as different biological replicates. **f** Leaf temperature of *cipk1-2* and *cipk1-3* mutants. The temperature of 9 leaves was measured in triplicate. **g** *CIPK1CA* drought sensitivity compared to the wild type. The data shown represent three independent experiments. **h** Water loss of *CIPK1CA* plant leaves. The experiment was repeated three times with independent treatments. Data are shown as means ± SD. **i** ABA-mediated stomatal closure in

*CIPK1CA* mutants. Stomatal width measurements are marked in the figure using red dashed lines. **j** Statistical analysis of ABA-induced stomatal closure in wild type, *CIPK1CA#1*, and *CIPK1CA#2* plants. The experiment was repeated three times as different biological replicates. **k** Leaf temperature of *CIPK1CA#1* and *CIPK1CA#2* plants. Leaf temperature was measured in nine leaves in triplicate. **l** *cbl1/9* and *cbl1/9/cipk1* drought resilience phenotype. (**m**) The leaf temperature of *cbl1/9* and *cbl1/9/cipk1* plants. Leaf temperature was measured in nine leaves in triplicate. Box plots show the 25% quantile, median (line), and 75% quantile. The whiskers plots (Tukey method) represent minimum and maximum values. Asterisks indicate statistical significance between the samples. Student's *t*-test (two-sided) (**c, h**), Two-way ANOVA analysis (**e, j**), One-way analysis (**f, k, m**) followed by Dunnett's multiple comparison test. Asterisk indicates significance compared to wild type: **$P$ < 0.01, ***$P$ < 0.001. Scale bars in (**a**) represent 20 μm, 2.0 cm in (**b, f, g, k, l, m**), and 10 μm in (**d, i**).

the interaction system and observed that the interaction between CIPK1 and PYLs only occurred at the cell membrane. This finding suggests that the interaction between CIPK1 and PYLs in plants is likely to occur at the cell membrane (Fig. 2b).

After establishing that CIPK1 can physically interact with ABA receptors, the split-luciferase complementation (LCI) assays were performed in *N. benthamiana*. In this analysis, CIPK1 was fused to the N-terminal fragment of *Photinus* luciferase (nLUC), while ABA

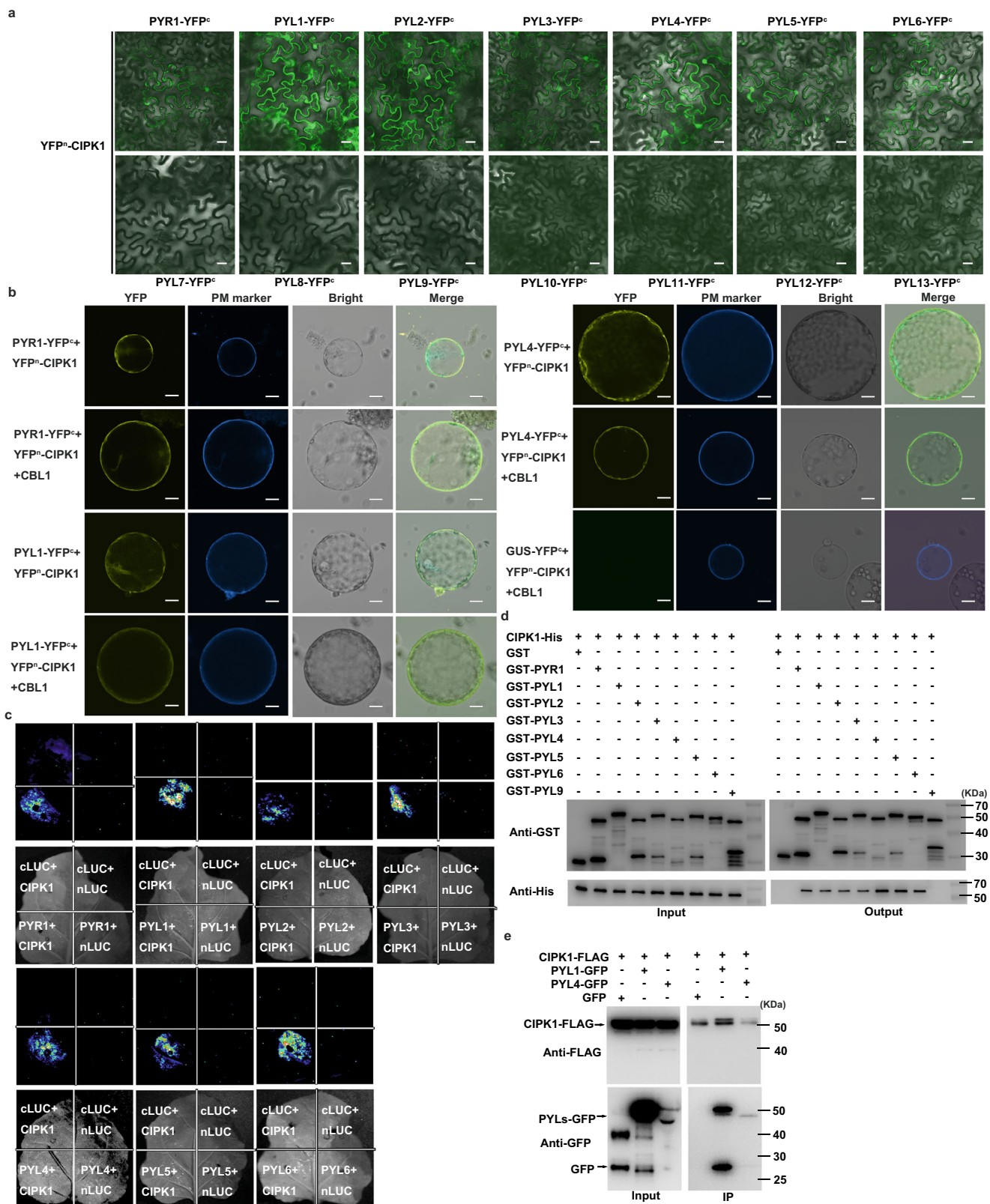

**Fig. 2 | CIPK1 physically interacts with PYR1/PYL1-6. a** CIPK1 interaction with PYR1/PYL1-PYL6 in BiFC assay. YFP^c, C-terminal portion of YFP. YFP^n, N-terminal portion of YFP. Scale bar, 40 µm. **b** CBL1 anchors the interaction of CIPK1 and PYL4 to the plasma membrane. Tobacco leaf protoplasts injected with different combinations were observed under confocal microscopy. CBL1n-OFP, in which the first 12 N-terminal amino acids of CBL1 are fused to OFP, was used as the plasma-

membrane marker (PM Marker). Scale bar, 20 µm. **c** CIPK1 interaction with PYR1/PYL1-PYL6 in LUC complementation imaging assays (LCI) assay. GUS-nLUC and GUS-cLUC were used as negative controls. **d** Pull-down assay showing CIPK1 interaction with PYR1/PYL1-PYL6. Anti-GST and Anti-His were used for protein detection. **e** Co-IP assay showing the interaction of CIPK1 with PYL1 and PYL4. The experiments were repeated three times (**a**–**e**) with similar results.

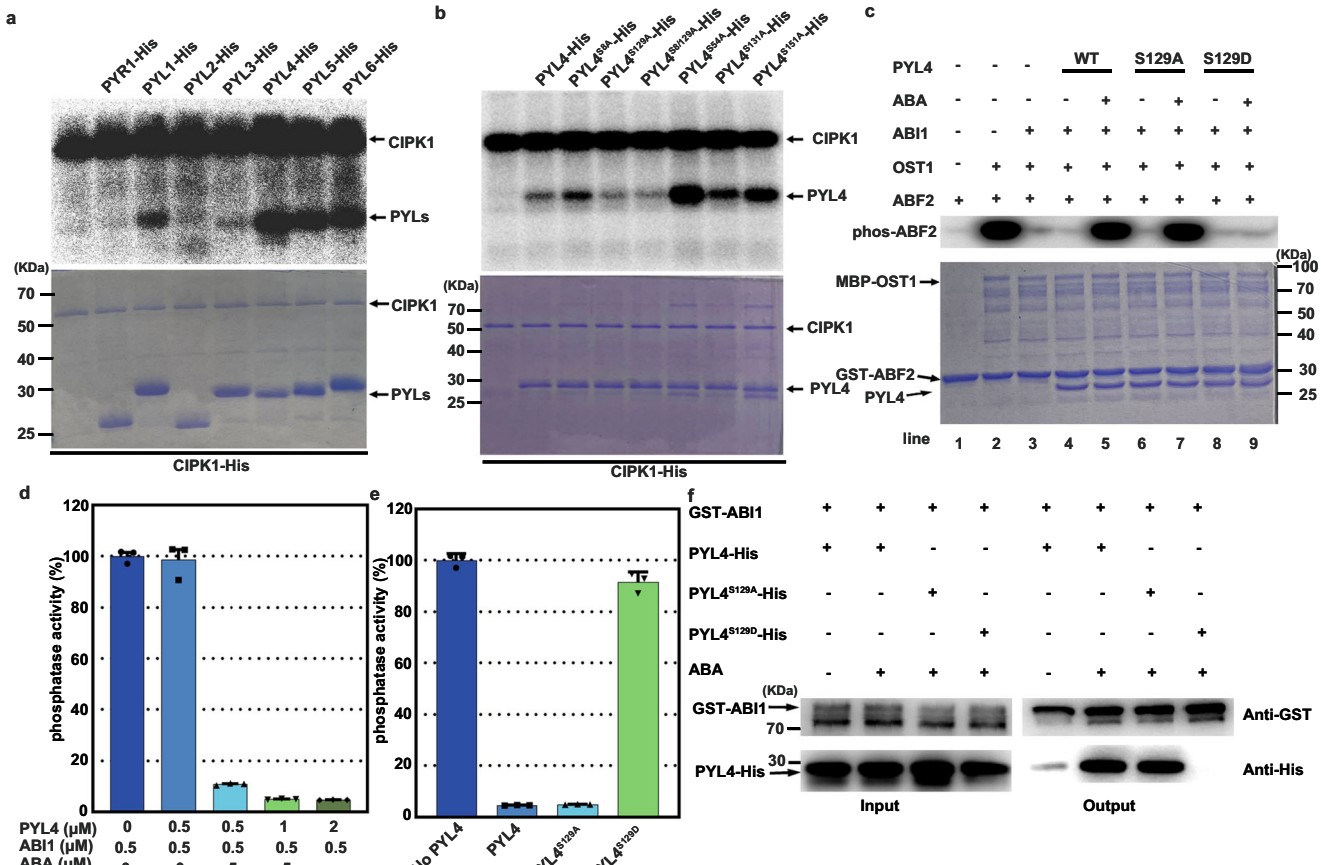

**Fig. 3 | CIPK1 phosphorylates PYLs, inhibiting their activities in vitro. a** CIPK1 phosphorylation of PYLs in vitro. In the assay, 1.5 μg CIPK1-His and ~2 μg PYLs-His were incubated for 30 min at 30 °C in a kinase buffer with 2 μCi [γ-$^{32}$P] ATP. The proteins were separated by 12% SDS-PAGE. Autoradiography (upper panel) and Coomassie staining (lower panel) show the phosphorylation and protein loading of the PYLs, respectively. The experiment was repeated two times with similar results. **b** Phosphorylation of PYL4 and its variants by CIPK1 in vitro. The phosphorylation experiment was repeated twice with similar results. **c** The phosphorylated form of

PYL4 is an inactive ABA receptor that could not inhibit the activities of ABI1 in the recombinant ABA signaling pathway in vitro. Autoradiography revealed phosphorylated ABF2. The experiment was repeated twice with similar results. **d** PYL4 inhibited the ABI1 phosphatase activities in the presence of ABA. Error bars, SEM ($n = 3$ independent experiments.). **e** Phosphomimic mutants of PYL4 did not inhibit ABI1 phosphatase activities. Error bars, SEM ($n = 3$ independent experiments.). **f** PYL4$^{S129D}$ did not interact with ABI1 in pull-down assays. The experiments were performed three times with similar results.

receptors and cLUC were combined to establish fusion proteins. After co-injection of CIPK1-nLUC and PYR1/PYL1-6-cLUC, recombinant LUC fluorescence activities were visible in tobacco leaves (Fig. 2c). In contrast, LUC fluorescence activities were not detected in the combination of CIPK1-nLUC and PYL7-PYL13-cLUC group and the corresponding negative control (Supplementary Fig. 2a, b). These results suggest that CIPK1 interacts with the PYR1/PYL1-6 ABA receptors.

Then, glutathione S-transferase pull-down (GST pull-down) assays were performed to confirm these interactions in vitro. In this assay, CIPK1 was fused to the 6xHis tag, while ABA receptors were fused to the GST tag to produce the fusion protein. Through pull-down assays using GST-PYL9 as a negative control, our results showed that CIPK1 only interacted with PYR1/PYL1-PYL6 (Fig. 2d). These findings imply that CIPK1 and ABA receptors PYR1/PYL1-PYL6 directly interacted in vitro.

To investigate whether CIPK1 physically interacts with ABA receptors in vivo, we extracted protein from transgenic *Arabidopsis* plants expressing both PYL1/4-GFP and CIPK1-FLAG. Empty GFP was used as a negative control. Our results showed that PYL1-GFP or PYL4-GFP were immunoprecipitated by CIPK1-FLAG (Fig. 2e). Taken together, these in vivo and in vitro assays demonstrate that CIPK1 physically interacts with ABA receptors.

## CIPK1 phosphorylates PYLs to inhibit their activities

One of the most prevalent biological regulatory mechanisms is phosphorylation by kinases to control the activities of targets. The CIPK-mediated phosphorylation events of downstream target proteins have been reported[47,48]. An in vitro phosphorylation assay in which PYR1/PYL1-6 was fused to His-tag and CIPK1-His was conducted as previously described[48]. In the assay to fuse PYLs with GST or MBP tags, they were both shown to be close to CIPK1-His protein bands. Meanwhile, the fusion of CIPK1 with other tags resulted in weak activities in our system. Therefore, we used CIPK1-His as the kinase and PYLs-His as the substrate for phosphorylation assays. Autoradiography revealed the phosphorylation band of ABA receptors by CIPK1, consistent with prior interaction results (Fig. 3a, and Supplementary Fig. 3a).

To establish the target of CIPK1 phosphorylation on ABA receptors, phosphorylation sites were predicted using the Group-based Prediction System web tool (http://gps.biocuckoo.cn/online.php). Then, several nonphosphorylated variants of PYL4, including PYL4$^{S8A}$, PYL4$^{S54A}$, PYL4$^{S129A}$, PYL4$^{S8/129A}$, PYL4$^{S131A}$, and PYL4$^{S151A}$ were generated. Phosphorylation intensities were significantly attenuated in the presence of Ser129 variants (Fig. 3b). By combining sequence conservation analysis and preferred phosphorylation pattern analysis of CIPK, Ser129 of PYL4 was shown to be evolutionarily conserved in ABA receptors (Supplementary Fig. 4), and was located close to the previously described potential ABA binding site (PYL4 Phe130)[50].

Protein kinases frequently modify the stability, activity, or localization of their substrates to affect downstream signaling[51,52]. Studies have explored the roles of PYLs mutations in ABA signaling[24,25,50]. First, in this study, the mechanisms via which phosphorylation affects PYL activities were evaluated using a modified in vitro system of the recombinant core signaling pathway (ABA-PYL-ABI1-OST1-ABF2), as previously described[24]. It was revealed that PYL4$^{S129A}$ inhibits ABI1 and promotes ABF2 phosphorylation similar to PYL4. However, PYL4$^{S129D}$ did not inhibit the phosphatase activities of ABI1 (Fig. 3c). Regarding the in vitro phosphatase activities, it was revealed that phosphorylated PYL1 and PYL4 diminished the ability to inhibit ABI1 phosphatase, consistent with findings of the in vitro recombinant core ABA signaling pathway (Fig. 3d, e, and Supplementary Fig. 3c). An analysis was conducted to determine whether the interactions between ABI1 and ABA receptors were affected by phosphomimic mutations at the conserved site in PYL4, using both pull-down and yeast two-hybrid assays. The results showed that the phosphomimic mutations reduced the ABI1-PYL interactions (Fig. 3f and Supplementary Fig. 3b, d). These results suggest that CIPK1-mediated phosphorylation of PYLs alleviated their interactions with ABI1.

## PYL4$^{Ser129}$ is essential for plant responses to ABA and drought stress

Next, the mechanisms via which PYL4$^{Ser129}$ phosphorylation impacts ABA responses in planta were investigated. The effects of phosphorylation on the localization of PYL4 were determined by constructing 35S promoter-driven PYL4, PYL4$^{S129A}$, or PYL4$^{S129D}$ fusion GFP transgenic plants. The phosphorylation did not alter the subcellular localization of PYL4 (Supplementary Fig. 5). Then, transgenic plants harboring native promoter-driven wild-type PYL4, PYL4$^{S129A}$, and PYL4$^{S129D}$ were created in quintuple-mutant pyr1pyl2pyl4pyl5pyl8 (abbreviated as 12458) plants (Fig. 4a). The ABA-hypersensitive phenotype of quintuple mutants was complemented in germination and seedling growth in transgenic plants of wild-type PYL4 or PYL4$^{S129A}$. In contrast, transgenic lines expressing PYL4$^{S129D}$ exhibited the ABA-insensitive phenotype of 12458 mutant plants (Fig. 4a, and Supplementary Fig. 6a, b). Meanwhile, treatment of these transgenic plants with ABA did not inhibit their growth. Although PYL4, PYL4$^{S129A}$, and PYL4$^{S129D}$ did not result in significant differences in root elongation when compared with 12458 (Supplementary Fig. 6c), PYL4 and PYL4$^{S129A}$ showed comparable shoot biomass to that of the wild type, while PYL4$^{S129D}$ and 12458 exhibited comparable shoot biomasses (Fig. 4b, and Supplementary Fig. 6d). Based on the in vitro findings and the results obtained from transgenic plants, it can be inferred that phosphomimetic mutated PYLs located at the conserved site corresponding to PYL4 Ser129 are not active receptors.

The effects of PYL4$^{S129}$ phosphorylation on plant growth and responses to stressful environments were also investigated. Since 12458 exhibits a highly drought-sensitive and impaired growth phenotype in normal conditions, the growth and development of these transgenic plants were tracked in the greenhouse (Fig. 4c). The transgenic PYL4$^{S129A}$ and PYL4 plants recovered the mutant growth defects and drought sensitivity (Fig. 4c, d). The drought-hypersensitive phenotypes of 12458 and PYL4$^{S129D}$ plants were found to be correlated with the rapid decrease in soil water content (Supplementary Fig. 1d). However, the phenotypes of PYL4$^{S129D}$ transgenic plants were comparable to those of 12458 mutants (Fig. 4c). The soil drought assay showed that PYL4 and PYL4$^{S129A}$ transgenic plants recovered the drought-sensitive phenotype of 12458, while PYL4$^{S129D}$ and 12458 mutants were more sensitive to drought (Fig. 4d). We conducted a study in which we statistically analyzed detached rosette leaves to determine their wilting status. Our results indicated that PYL4$^{S129D}$ had similar water transpiration rates to 12458 but at a higher level (Fig. 4e). This was in contrast to PYL4 and PYL4$^{S129A}$, which exhibited similar water transpiration rates to the wild type (Fig. 4e). We also carried out a leaf

temperature analysis, which showed that PYL4 and PYL4$^{S129A}$ had a comparable water transpiration rate to the wild type (Fig. 4f). However, PYL4$^{S129D}$ exhibited lower leaf temperature and higher water transpiration rates than the wild type, comparable to 12458. These findings indicate that PYL4 Ser129 is crucial in responses to ABA in plants.

## ABA directly inhibits CIPK1-mediated phosphorylation of PYLs

The major component of ABA signaling, SnRK2s, interacts with CIPKs[53] and OST1 regulation of TOR kinase activities by phosphorylation plays a key role in modulating stress responses and growth[24]. In this study, the LCI assay showed that there were no interactions between CIPK1 and OST1 (Supplemental Fig. 2c). Subsequently, we conducted pull-down and Co-IP assays to examine the impact of ABA on the interaction between CIPK1 and PYL1, as well as PYL4. Our findings indicated that the presence of ABA decreased the binding strength between CIPK1 and, PYL1 and PYL4 (Fig. 5a–c and Supplementary Fig. 7). Consistently, the phosphorylation of PYL4 was partially inhibited by elevated ABA concentrations (Fig. 5d). Moreover, supplementation of ABA did not alter the kinase activities of CIPK1. Therefore, ABA directly inhibits the interactions between CIPK1 and PYL4.

To establish whether phosphorylation exerts regulatory functions on PYLs in plants in response to ABA, we assessed whether exogenous ABA affects the phosphorylation of PYL4. Transgenic plants over-expressing CIPK1-FLAG were grown for 24 h under 60 µM ABA treatment. Then, CIPK1-FLAG was extracted, enriched with anti-FLAG antibodies, and incubated at 30 °C for 30 min with recombinant PYL4-His and 2 µCi [γ-$^{32}$P] ATP (Fig. 5e). It was established that within 6 h, ABA treatment did not alter CIPK1-mediated phosphorylation of PYL4, but from the 9th h, it significantly suppressed PYL4 phosphorylation. These findings suggest that ABA can directly inhibit CIPK1-mediated phosphorylation of PYL4.

To determine whether Ca$^{2+}$ plays a vital role in CIPK1-mediated phosphorylation of PYLs, we performed in vitro phosphorylation assays. Our results revealed that the presence of EGTA, a Ca$^{2+}$ chelator, in the reaction significantly impacted the phosphorylation of PYL4 by CIPK1. Furthermore, we found that the addition of Ca$^{2+}$ to the reaction had no significant effect on CIPK1 autophosphorylation at high concentrations of Ca$^{2+}$. However, it did affect the phosphorylation of PYL4 by CIPK1, indicating that PYL4 is likely to be phosphorylated by CIPK1 at lower cytosolic Ca$^{2+}$ levels (Fig. 5f).

To further elucidate if CIPK1-mediated phosphorylation of PYLs requires CBL1/9, an in vivo phosphorylation assay was conducted. We subjected PYL4, PYL4$^{S129A,}$ and PYL4-FLAG/cbl1/9 transgenic plants to ABA treatment for 0, 6, and 9 hours. The PYL4 protein was then extracted and isolated using anti-FLAG antibodies, and the phosphorylation level of PYL4 was determined via anti-P-Ser antibodies. Notably, significant phosphorylation bands could be detected in PYL4 plants, while the intensity of phosphorylation was significantly depleted at the 9th h. In addition, only weak phosphorylation bands were observed in PYL4$^{S129A}$ and PYL4-FLAG/cbl1/9 plants compared with PYL4 plants; the phosphorylation bands in PYL4-FLAG/cbl1/9 plants were stronger than those in PYL4$^{S129A}$ plants, suggesting that there may be other CBLs involved in the phosphorylation of PYL4 by CIPK1 in vivo (Fig. 5g). These results unequivocally established that CIPK1-mediated phosphorylation of PYL4 requires CBL1/9.

To verify their genetic interactions, the 12458/cipk1 sextuple mutant was obtained by hybridization. The 12458/cbl1/9 heptamutant was not generated by hybridization because CBL1 and PYR1 are localized in close positions on chromosome 4, and CBL9 is positioned in the same chromosome with both PYL5 and PYL8 in Arabidopsis. Then, a CRISPR-Cas9 vector was constructed to knock out CBL1 and CBL9 in 12458 plants; however, the 12458/cbl1/9 heptamutant was not obtained. Comparisons of water loss among WT, cipk1-3, 12458, and 12458/cipk1 plants in soil revealed comparable drought sensitivities between 12458 and 12458/cipk1 with the same soil water stress (Fig. 5h, and

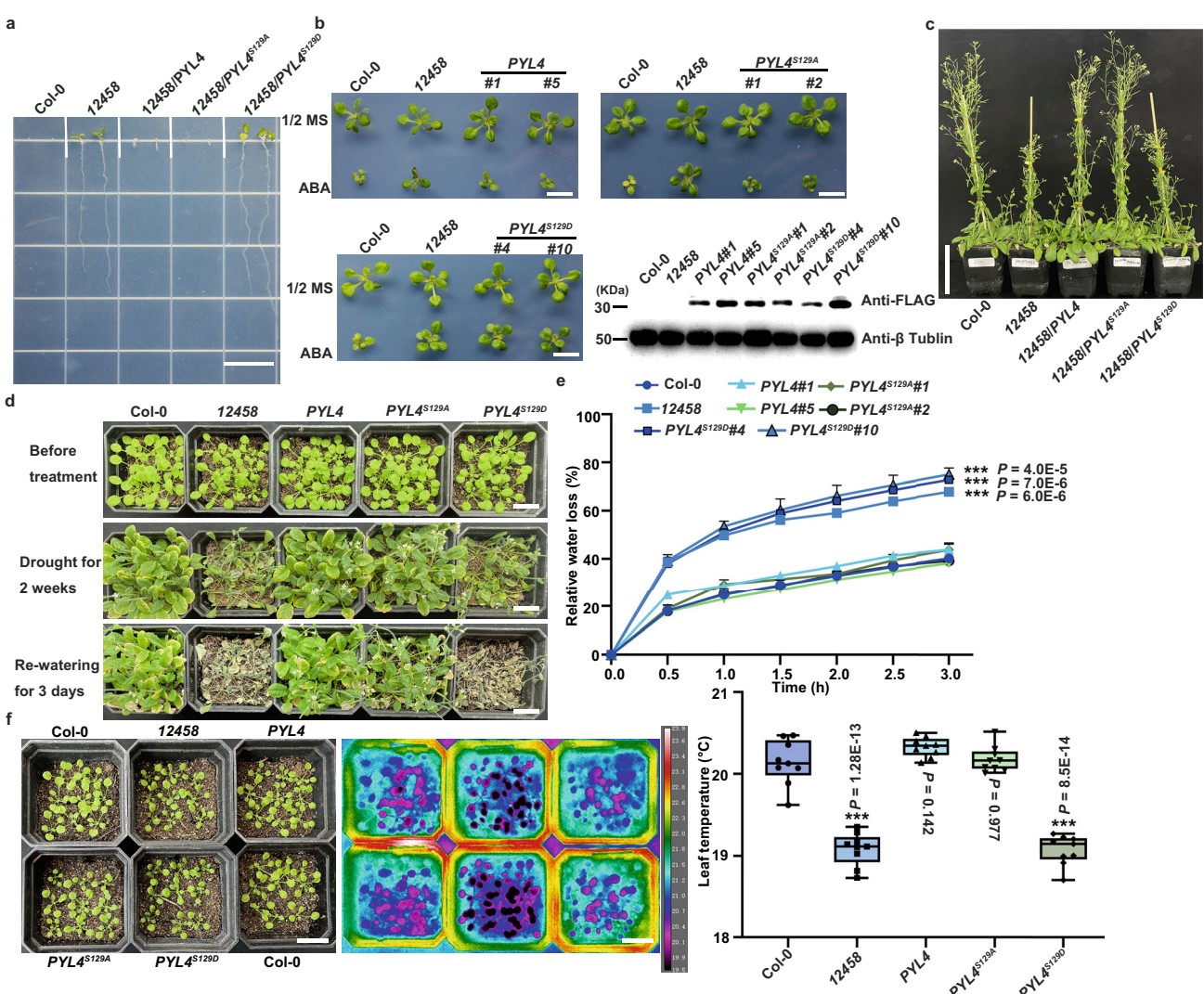

**Fig. 4 | Functional analysis of PYL4 phosphorylated by CIPK1 in vivo.**
**a** Germinating seeds were grown on 1/2 Murashige-Skoog (MS) medium containing 3 μM ABA and imaged after 5 d. Scale bar, 1.4 cm. The experiments was repeated thrice with similar results. **b** Images of 14-day-old seedlings growing on 1/2 MS with or without 20 μM ABA. Protein levels of transgenic materials used in Fig. 4. The PYL4 protein was detected using Anti-FLAG antibodies. Scale bar, 1.4 cm. Three independent replications of the experiment yielded similar results. **c** Images of *PYL4* transgenic plants growing normally for 40–50 d in the greenhouse. Scale bar, 8.0 cm. Three independent replicate experiments obtained similar results. (**d**) Phenotype of *PYL4/12458*, *PYL4^S129A/12458*, and *PYL4^S129D/12458* under drought conditions. Scale bar, 2.0 cm. The experiment was repeated three times with similar results. **e** water loss measure in detached leaves of *PYL4* transgenic plants. The experiment was repeated three times with independent treatments. Data are means ± SD of 3 replicates. Asterisks indicate significance compared to the wild type. ***$P < 0.001$, Student's *t*-test (two-sided). **f** Leaf temperature *of PYL4* transgenic plants. Leaf temperature was measured on nine leaves in triplicates. Scale bar, 2.0 cm. Box plots show the 25% quantile, median (line), and 75% quantile. The whiskers plots (Tukey method) represent minimum and maximum values. Statistical analysis was performed by one-way ANOVA analysis followed by Dunnett's multiple comparison test. Asterisks indicate significance compared to the wild type (**$P < 0.01$).

Supplementary Fig. 1e). The leaf temperatures of different plants were also measured, and the results showed that *12458/cipk1* and *12458* had similar water transpiration (Fig. 5i). In summary, CIPK1 negatively regulates *Arabidopsis* responses to drought by suppressing ABA receptor activities via phosphorylation. This confirms the existence and physiological functions of the Ca²⁺-CBL1/9-CIPK1-PYLs signaling pathway, which is probably involved in the negative feedback of Ca²⁺ on ABA signaling (Fig. 6).

## Discussion

The regulatory system that balances plant growth and stress responses largely depend on ABA signaling[54]. This regulatory network is regulated during growth and stress responses[55–57]. The changing external environment causes Ca²⁺ oscillations in plant cells, and intracellular Ca²⁺ sensors directly induce downstream signaling networks to regulate plant growth and stress responses[31]. In this study, we reveal a mechanism whereby CBL1/9-CIPK1 phosphorylation inhibits ABA signaling. The CBL1/9-activated CPK1 directly phosphorylates PYLs to negatively regulate drought stress. Additionally, under normal growth conditions, CIPK1-mediated phosphorylation of PYLs at Ser129 of PYL4 inactivates PYLs that fail to inhibit PP2C, such as ABI1. When plants are exposed to drought stress or ABA treatment, ABA inhibits PYLs phosphorylation by directly blocking the interactions between CIPK1 and PYLs. This ensures plant survival in harsh conditions. Therefore, the CBL1/9-CIPK1 complex appears to be a key mediator of stress response by integrating the Ca²⁺ and ABA signaling pathways. Overall, these findings provide insights into the complex signaling networks that regulate plant responses to drought stress, which could be useful in developing strategies to improve crop resilience to water scarcity.

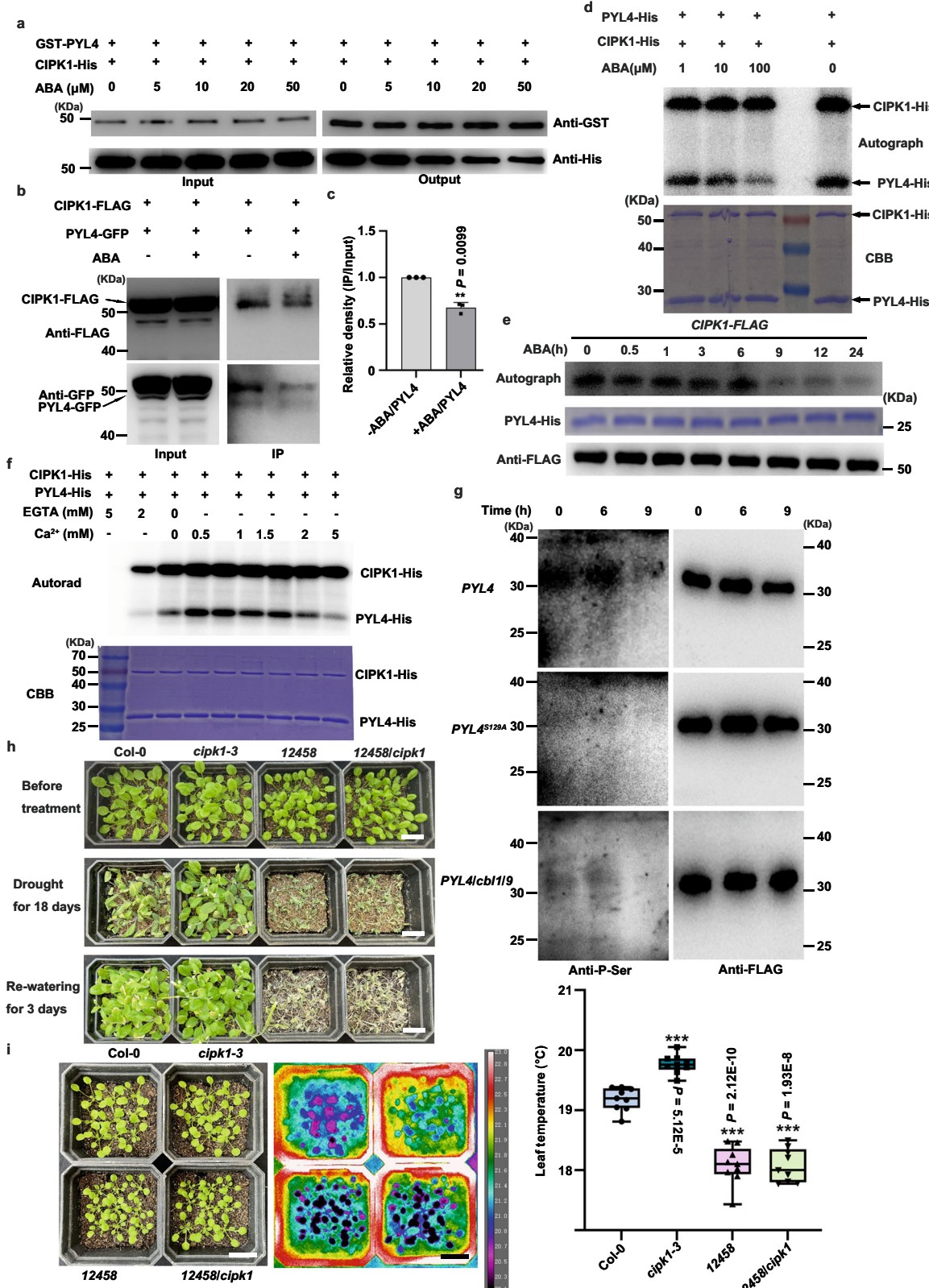

## CBL1/9-CIPK1 negatively regulates ABA responses during drought

The second messenger $Ca^{2+}$ signaling regulates various aspects of eukaryotic physiology[31]. In response to several stimuli, specific calcium signatures are created in plants via the influx of extracellular $Ca^{2+}$ or the release of $Ca^{2+}$ in organelles[58]. The different $Ca^{2+}$ signals are sensed by different calcium sensors to activate specific physiological responses. In plants, calcium sensors (CBLs) interact with specific CIPKs to regulate diverse physiological processes[36]. In vivo, CBL1/9 and CIPK1 exhibit comparable expression patterns, and CBL1/9 recruits CIPK1 to the plasma membrane[38]. The loss of *cbl1* does not alter the sensitivity of plants to ABA[38], whereas CBL9 responds to ABA

**Fig. 5 | Genetic interactions among CIPK1 with PYLs. a** ABA treatment inhibits the interaction between CIPK1 and PYL4 in pull-down assays. The experiment was conducted three times, and similar results were obtained. **b** ABA inhibits the interaction between CIPK1 and PYL4 in Co-IP assays. For ABA treatment, the extracted protein containing 100 μM ABA was incubated overnight with Anti-FLAG antibodies. Three repetitions of the experiment obtained similar results. **c** The relative density of PYL4 (IP/Input) in (**b**) reveals an interaction between CIPK1 and PYL4 in vivo that was reduced by ABA. Data were measured using ImageJ and represented as IP/Input. Data are shown as means ± SD. Asterisks indicate significance compared to the IP without ABA according to Student's *t*-test (Two-sided, ** $P < 0.01$). **d** ABA inhibits the phosphorylation of PYL4 by CIPK1 in vitro. In this assay, indicated concentrations of ABA was added to the reaction system. Two repetitions of the experiment yielded similar results. **e** Protein kinase assay of CIPK1 with PYL4-His under ABA treatment. Protein kinases were quantified by western blot and shown at the bottom. The experiment was repeated twice and similar

results were obtained. **f** Phosphorylation of PYL4 by CIPK1 requires a moderate amount of $Ca^{2+}$. The experiment was repeated three times with similar results. **g** CBL1/9-CIPK1 mediates the phosphorylation of PYL4 Ser129 in vivo. The phosphorylation signal of PYL4 was detected by anti-P-Ser antibodies, and the proteins were quantified through western blot and anti-FLAG antibodies. The experiment was conducted two times with similar results. **h** The *12458/cipk1* mutants showed similar drought sensitivity. Scale bar, 2.0 cm. The experiment was repeated three times with similar results. **i** *12458* and *12458/cipk1* plants leave temperature imaging. Data were statistical for each replicate of 9 leaves repeated three times. Box plots show the 25% quantile, median (line), and 75% quantile. The whiskers plots (Tukey method) represent minimum and maximum values. Statistical analysis was performed by one-way ANOVA analysis followed by Dunnett's multiple comparison test. Asterisk indicates significance compared to wild type: ** $P < 0.01$. Scale bar, 2.0 cm.

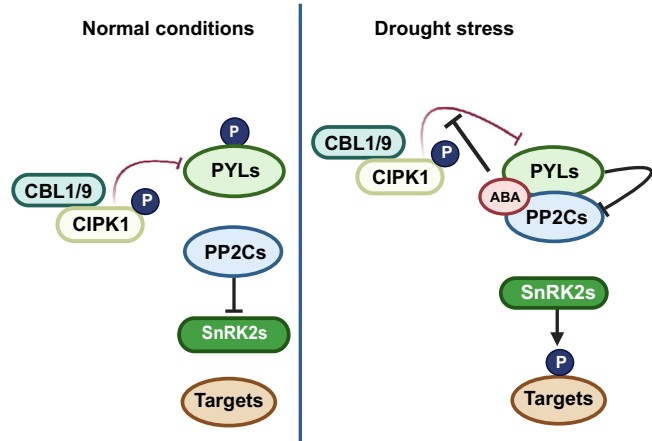

**Fig. 6 | A working model of CBL1/9-CIPK1-PYLs module regulating plant drought stress responses.** A proposed model illustrating how the phosphorylation of ABA receptors by CIPK1 acts as a negative regulator of drought stress responses in plants. Under normal conditions, CIPK1 acts to suppress the activation of ABA responses by phosphorylating PYL4 at Ser129, which inactivates the ABA receptor and prevents it from inhibiting the phosphatase activities of ABI1/PP2Cs. This results in a lack of downstream ABA signaling. However, under drought stress conditions, ABA inhibits CBL1/9-CIPK1-mediated phosphorylation of PYR1/PYL1-6, and active ABA receptors interact with ABI1/PP2Cs to inhibit their phosphatase activity, leading to the activation of downstream ABA signaling. This response enables plants to survive under drought stress. Positive regulation is indicated by arrows with solid lines, while strong negative regulation is represented by a solid line with a horizontal bar.

signaling[59]. Even though *cbl1* is sensitive to drought[60], the *cbl1/9* double mutant exhibits drought resilience, while still being ABA-sensitive in stomatal movement[40]. In this study, it was established that *cipk1* has drought resilience, whereas stomata exhibit hypersensitive responses to ABA. In *Arabidopsis*, CIPK17 and CIPK1 are close homologs. Although CIPK17 has also been noted to inhibit ABA responses[41], consistent with our results, CIPK1 is primarily expressed in shoots, whereas CIPK17 is primarily expressed in roots[61]. Preferentially, CIPK17 binds CBL2/3 at the tonoplast membrane to regulate stomatal movement[41], while CIPK1 interacts with CBL1/9 to function at the plasma membrane[38]. Moreover, *cbl1/9* and *cipk23* mutants exhibit drought resilience and ABA hypersensitivity outcomes in stomata, while CBL1/9-CIPK23 phosphorylates and activates SLAC1[40,45]. How does this positive regulator of SLAC1 negatively regulate ABA-induced stomatal closure? In addition to the inhibition of SLAC1-type ion channels by CIPK23[T190D] in guard cells[62]. Preferentially, CBL1/9 negatively regulates drought stress by activating CIPK1 to phosphorylate ABA receptors.

## CIPK1 interacts with and phosphorylates PYLs to abolish their activities

In the past two decades, the discovery of the ABA receptor has been the most significant outcome in studies on ABA perception and signaling[6,7]. Physiologically, PYLs are regulated at the transcriptional and posttranscriptional levels. During seed germination, ABA inhibits PYLs expressions whereas ABI5 positively regulates PYLs expressions to regulate seed germination[63]. In plants, PYLs also undergo post-transcriptional modifications, including tyrosine nitration, phosphorylation, and ubiquitination. These adjustments regulate the fine-tuning of responses to environmental changes. Regarding the molecular mechanism via which CIPK1 regulates ABA signaling, interaction analyses were performed to investigate whether CIPK1 directly physically interacts with and regulates PYLs. The interaction analyses identified the ABA receptor (PYR1/PYL1/2/3/4/5/6) to be an important target that is regulated by CIPK1.

Moreover, CIPK1 was found to primarily phosphorylate PYL4 at Ser129. Biochemical analyses showed that phosphorylation regulates ABA receptor functions and that the phosphomimicking PYL1 and PYL4 versions failed to inhibit ABI1 phosphatase activities in the presence of ABA. Our complementary assays showed that *PYL4* and *PYL4$^{S129A}$* can significantly restore ABA insensitivity of *12458* during seed germination and seedling growth. When 4-day-old seedlings were grown on an ABA-supplemented medium, there were no significant differences in root elongation among the mutants. This may be because *1258* has significant ABA insensitivity and *PYL4* is mainly expressed in seeds and stomata[5]. The *PYL4* and *PYL4$^{S129A}$* complementary transgenic lines almost completely restored the growth inhibitory characteristics and drought sensitivity of *12458*. The phosphomimic *PYL4$^{S129D}$* showed a comparable phenotype to *12485*. These results suggest that CIPK1 phosphorylates the ABA receptor and inhibits the activities of PYLs.

## High concentration of ABA inhibits the phosphorylation of PYLs by CBL1/9-CIPK1

Functionally, CBL anchors CIPK to a specific location. The CBL-CIPK complex plays an important regulatory role in plant growth, development, and responses to the external environment[31,64]. Both CBL1/9 and CIPK1 are involved in various processes in *Arabidopsis*, including potassium absorption and drought stress responses[48,65]. This study established that CBL1/9-CIPK1 is a negative regulator of ABA signaling, thus, the CBL1/9-CIPK1 complex may be a key node in plant responses to stress and growth. CBL1/9-CIPK1 serves as a regulatory mechanism that inhibits the occurrence of aberrant ABA signaling in plants. This is achieved through the phosphorylation of PYLs at lower concentrations of ABA or reduced levels of $Ca^{2+}$ concentration.

More than 1 μM ABA concentration was shown to inhibit the growth of plant roots[66]. However, low ABA levels (2 nM) in *aba2-1* protoplast were sufficient for activating ABA responses[67]. Structural

studies highlight the conserved gate-latch-lock mechanism of ABA perception and signal transduction[68–72]. When the ABA receptor binds ABA, the conformational change allows the ligand to entry the gate onto the latch, allowing the receptor to dock with PP2Cs. A tryptophan conserved in the PP2C active site is in turn inserted between the gate and the latch, further locking the receptor-PP2C complex[68]. The capacity of PYLs to bind ABA is nearly 100-fold higher when clade A PP2C proteins are present[73]. Therefore, PP2Cs can be considered as co-receptors for ABA, which have an important role in the stabilization of ABA signaling and ABA-PYLs-PP2Cs complexes in plants[7]. The phosphorylated PYLs cannot bind to PP2C, which is detrimental to the binding of ABA and ABA signaling.

Additionally, previous study has shown that ABA could inhibit the interaction and phosphorylation of PYLs by EL1-like casein kinases to prevent the degradation of ABA receptors[25]. In this study, we found that ABA impaired the interactions between CIPK1 and PYLs, and phosphorylation activities of PYL4 by CIPK1 in plants decreased significantly from 9 h after 60 μM ABA treatment. Phosphorylated PYL4 failed to respond to ABA signaling and inhibited the phosphatase activities of ABI1. These results also suggested that CIPK1-mediated phosphorylation of PYLs may occur under normal conditions or in low ABA concentrations. These phosphorylated PYLs cannot activate ABA signals. When subjected to external stress, high concentrations of ABA in plants directly suppress CIPK1 phosphorylation of PYLs to ensure ABA responses and allow plants to survive the harsh environment.

## Methods

### Plant materials and growth conditions

The Columbia (Col-0) accession was used as the control for all experiments. The T-DNA insertion lines (*cipk1-2* (SAIL_521_F05), *cipk1-3* (GK230-D11), and *pyr1pyl2pyl4pyl5pyl8*) were obtained from the Nottingham *Arabidopsis* Stock Centre (NASC). The T-DNA insertion line (*cbl1/9*) was described previously[48]. The *pyr1pyl2pyl4pyl5pyl8/cipk1* sextuple mutant was generated by hybridizing *pyr1pyl2pyl4pyl5pyl8* with *cipk1-3* while the *cbl1/9/cipk1* triple mutant was generated by hybridizing *cbl1/9* with *cipk1-2*.

To generate overexpression or complementary transgenic plants, each different CDS (coding sequence) fused FLAG/GFP tag driven by 35 S or native promoter was cloned into a pCAMBIA1300 vector. All constructs were transduced into *Agrobacterium* cells (GV3101), which were grown at 28 °C for 2 d, and then transformed into the wild type or related mutants. The primers used in this study are listed in Supplementary Data 1.

*Arabidopsis* seeds were surface sterilized for 15 min in 10% bleach, rinsed six times in sterile-deionized water, sown on 1/2 MS medium (PhytoTech) supplemented with 1% (w/v) sucrose and 0.6% (w/v) (for horizontal growth) or 1% (w/v) (for vertical growth) agar and stratified for 3 d at 4 °C. Then, the plates were transferred to a growth chamber with a long-day cycle (16 h light/8 h dark) at 22 °C. For soil cultures, the 7-day-old vertical growth seedlings were transferred to nutrient-rich soil (Pindstrup substrate, Denmark) and grown in a greenhouse under controlled conditions (22 °C, 16 h light/8 h dark, a regime with light intensity adjusted to 120 μmol m$^{-2}$ s$^{-1}$).

### The GUS Histochemical assay

Leaves of 4-week-old transgenic plants expressing a GUS reporter gene under *CIPK1* native promoter were cut off and immersed in a staining buffer at 37 °C for 12 h. The samples were washed using 70% ethanol after which epidermis images were microscopically (OLYMPUS DP80) obtained.

### Physiological experiments

For the soil drought assay, nine 7-day-old seedlings were placed each in a pot with the same amount of soil and grown for another 7 d in the greenhouse. After each pot had fully absorbed the water, excess water was removed. Then, water was withheld from these plants for about 2-3 weeks, and during this period, the pot was regularly replaced to avoid the position effects. When differential drought phenotype appeared, images were obtained after which the pots were re-watered for 2-3 d and imaged again.

For the ABA-induced stomatal closure assay, the assay was performed as previously described[74]. In brief, the 5th or 6th rosette leaves of *Arabidopsis thaliana* that had been grown for about 20 d in the greenhouse were obtained, placed in MES buffer (10 mM MES-KOH, pH 6.15, 10 mM KCl, and 50 μM CaCl$_2$) and incubated in a light incubator for 2.5 h to fully open the stomata. Then, the leaves were transferred to a MES buffer containing 10 μM ABA and incubated for 2 h. The epidermal strips were microscopically (OLYMPUS DP80) imaged. The minimum pore space between the two guard cells that make up the stomata was the stomata width. Finally, the stomatal apertures were measured using the Image J software. The data for each ecotype was collected from three repetitions, with a total of no less than 100 stomata measured.

For the water loss of detached leaves, five seedlings per pot in soil were grown in the greenhouse for 3-4 weeks, and the rosette leaves were cut and placed on a piece of weighing paper. The rosette leaves were periodically weighed for 3 h. Water loss was shown as a percentage of the original fresh weight. At least three different experiments were performed.

For infrared thermography assay, about 3-week-old seedlings grown in the greenhouse had water withheld for 6-8 days (soil moisture 70%-85%) before being imaged with the FLIR A655sc infrared camera (Teledyne FLIR). Leaf temperature was measured using FLIR ResearchIR Max professional software.

For the germination assays, seeds for each genotype were sown in 1/2 MS medium containing 3 μM ABA, and incubated in a light incubator for 3 d after which radicle emergence was analyzed.

For the growth assays, 3-4-day-old vertically cultured seedlings were transferred to 1/2 MS medium containing 20 μM ABA, and incubated in a light incubator for 10 d. The fresh weights were measured on the 10th d.

### The RNA expression assay

Total RNA was extracted using the RNAsimple Total RNA Kit (DP419, TIANGEN). Then, 2 μg RNA was used for first-strand cDNA synthesis with the HiScript II Q RT SuperMix for qPCR (+gDNA wiper) (R223, vazyme). *NtActin* was used as the internal control. The primers used for RT-PCR are shown in Supplementary Data 1.

### The bimolecular fluorescence complementation (BiFC) assay

The BiFC assay was conducted as previously described[75]. Briefly, *Agrobacterium* was resuspended in a buffer (10 mM MES, 10 mM MgCl$_2$, pH = 5.6) and mixed to a specific concentration. The YFP$^n$-CIPK1 and PYR1/PYLs-YFP$^c$ were transformed into *N. benthamiana* leaves and expressed for 4 d. Fluorescence signals were captured using a confocal laser scanning microscope (Olympus IX83-FV3000). Tobacco protoplasts BiFC assay was conducted as previously described[47].

### The split-luciferase complementation (LCI) assay

The LCI assay was conducted as previously described[75]. Briefly, the resuspension method was consistent with that of the BiFC assay. In this assay, CIPK1-nLUC and PYR1/PYLs-cLUC were transformed into *Agrobacterium* GV3101 and co-injected into *N. benthamiana* leaves. After 72 h, signals were detected by a cold charge-coupled device camera (CCD; Lumazone Pylon 2048B; Princeton).

### In vitro pull-down assay

The in vitro pull-down assay was performed as previously described[76]. Briefly, 5 μg of purified GST-PYR1/PYLs recombination protein was

incubated with GST-Sefinose (TM) Resin 4FF (Settled Resin) (BBI) at 4 °C for 2 h and incubated with 2 μg of CIPK1-His for another 2 h in a pull-down buffer, after which the beads were washed 6 times using the pull-down buffer. The proteins were separated by SDS-PAGE for the immunoblot assay using anti-His (1:5000, TransGen, HT501) antibody and anti-GST (1:5000, TransGen, HT601) antibody. The interactions between ABI1 and PYLs were also determined in pull-down assays.

For ABA inhibition of CIPK1 interaction with PYLs, 5 μg of CIPK1-His protein was incubated on Ni-NTA beads for 2 h and incubated with 2 μg of GST-PYL1 or GST-PYL4 for another 2 h in the pull-down buffer. Then the beads were washed 6 times using the pull-down buffer. The proteins were separated by SDS-PAGE for the immunoblot assay using anti-His (1:5000, TransGen, HT501) antibody and anti-GST (1:5000, TransGen, HT601) antibody.

## Co-immunoprecipitation (Co-IP) assay

The Co-IP assay was performed as previously described[47]. The constructs were co-expressed in transgenic *Arabidopsis*. The CIPK1 and PYL1/PYL4 were respectively detected using anti-FLAG antibodies (1:5000, TransGen, HT201) and anti-GFP antibodies (1:5000, TransGen, HT801). For ABA inhibits the interaction between CIPK1 and PYL4 in Co-IP assays, 100 μM ABA was added to the protein extracts. Then the Co-IP assay was performed as previously described[47].

## Protein kinase assays

The PYR1-His, PYL1-His, PYL2-His, PYL3-His, PYL4-His, PYL5-His, PYL6-His, CIPK1-His, MBP-OST1, GST-ABI1 and GST-ABF2 recombinant protein was purified from *E. coli* BL21 (Rosetta). Then, the in vitro protein kinase assays were performed as previously described[24,47]. For CIPK1 and PYLs, in the assay, 1.5 μg CIPK1-His and ∼ 2 μg PYLs-His were incubated for 30 min at 30 °C in a kinase buffer with 2 μCi [γ-$^{32}$P] ATP. For in vitro recombinant core ABA signaling pathway, about 2 μg wild-type or mutated PYL4 proteins were incubated with 0.5 μg GST-ABI1 with or without 5 μM ABA in 20 μL of reaction buffer, the reactions without PYL4 were used as control. After 15 minutes incubation, 1 μg MBP-SnRK2.6 was added to the reaction and incubated for additional 15 minutes. Then, a mixture containing 1 μM ATP, 1 μCi [γ-$^{32}$P] ATP, 2 μg GST-ABF2 fragment, and phosphatase inhibit cocktail 3 was add to the reaction to total volume of 25 μL. The reaction mixtures were incubated for 30 minutes at 30 °C. The proteins were separated by 12% SDS-PAGE. Autoradiography (upper panel) and Coomassie staining (lower panel) show the phosphorylation and protein loading of the PYLs, respectively. The phosphorylation signals are detected using a Typhoon 9410 imager (Bio-Rad).

For the in vivo protein kinase assay, *CaMV35S: CIPK1-FLAG* transgenic seedlings were grown on 1/2 MS medium for 10 days. The seedlings were transferred to plates containing 60 μM ABA and treated for different times (0, 0.5, 1, 3, 6, 9, 12, 24 h), then the proteins were extracted using IP buffer and incubated with anti-FLAG agarose (Sigma-Aldrich, #A2220) for 2 h at 4 °C. Then, samples were eluted with 40 μL 3× FLAG peptide (Sigma, #F4799) for another 2 h at 4 °C. The CIPK1-FLAG protein was incubated with PYL4-His in the kinase reaction buffer at 30 °C for 30 min. The subsequent procedures were consistent with those of the in vitro phosphorylation assay.

For the in vivo protein kinase assay, PYL4 protein was immunoprecipitated with anti-FLAG antibodies in *PYL4, PYL4-FLAG/cbl1/9*, and *PYL4$^{S129A}$* plants. The PYL4 signals were detected with anti-FLAG antibodies (1:5000, TransGen, HT201), and the phosphorylation signals were analyzed by western blotting using anti-phospho-Ser antibodies (1:2000, ImmunoWay, #YM3440).

## In vitro phosphatase activity assay

For PYL1 and PYL4, the Serine/Threonine Phosphatase Assay System (Promega) was used to determine the PP2C activities as previously described[77]. Briefly, recombinant proteins (PYL1-His, PYL1$^{T133A}$-His,

PYL1$^{T133D}$-His, PYL4-His, PYL4$^{S129A}$-His, PYL4$^{S129D}$-His, and GST-ABI1) were incubated at room temperature for 15 min after which the samples were mixed with the reaction buffer (250 mM imidazole (pH7.2), 1 mM EGTA, 25 mM MgCl$_2$, 0.1% β-mercaptoethanol, and 0.5 mg/mL BSA) and the synthetic phosphopeptide substrate (RRA[pT]VA). The mixture was incubated at 37 °C for 30 min and supplemented with 50 μL Molybdate dye/additive mixture to terminate the reaction. After incubation at room temperature for 20 min, the color was developed and the absorbance was measured at 600 nm using a BioTek microplate spectrophotometer. The amount of phosphate released from the sample was determined according to the standard curve prepared using the 0, 100, 200, 500, 1000, and 2000 pmol phosphate solution.

## The yeast two-hybrid assay

For the yeast two-hybrid assay, the CDS of *ABI1* was cloned into the pGADT7 vector while the CDS of *PYL4* was cloned into a pGBKT7 vector. Then, they were used for site-directed mutagenesis using specific primers (Supplementary Data 1). To investigate protein interactions between ABI1 and PYL4, the plasmids were co-transformed into *Saccharomyces cerevisiae* AH109 cells. After 2 d of incubation on yeast SD/-Leu/-Trp at 30 °C, the SD/-Leu/-Trp liquid medium was used to culture yeasts to an optical density (OD = 1.0). To determine the intensity of protein interactions, saturated yeast cultures were diluted to 10$^{-1}$, 10$^{-2}$, and 10$^{-3}$ after which 3 μL of each dilution was spotted onto SD/-Leu/-Trp/-His medium in the absence or presence of 10 μM ABA and cultured at 30 °C for 4 d.

## Analysis of subcellular localization of PYL4

The PYL4-GFP transgenic plants were vertically grown on 1/2 MS medium. Then, 5-day-old seedlings were treated with 60 μM ABA for 1 h after which GFP fluorescence was observed using a confocal microscope (IX83-FV3000; Olympus).

## Sequence alignment and conservation analysis

To obtain different PYLs of different species, we used *Arabidopsis* PYLs to obtain possible PYLs of 12 different species by BLAST in NCBI. A total of 122 candidate proteins whose percentage was not less than 50% were retained. Sequence alignment was performed using the Jalview software. Amino acid conservativeness analysis was performed using the MEME (https://meme-suite.org/meme/tools/meme) online tool.

## Statistics and reproducibility

The *Arabidopsis* plants used in this study were grown in a greenhouse or incubator at 16 / 8 h (day/night) and 22 °C. To avoid bias, the relative soil water content was ensured to be consistent before the start of the drought experiments and the position of the small pots was changed regularly every day during the experiments; the stomatal apertures were counted as randomized statistics for the same ecotype with a sufficient sample size. For microscopic and physiological experiments, as least three completely independent experiments were performed. For phosphorylation assays, each experiment was repeated at least twice. No data were excluded from the study. Source data underlying the figures are provided as a Source Data file.

## Reporting summary

Further information on research design is available in the Nature Portfolio Reporting Summary linked to this article.

## Data availability

All data generated and analyzed in this study are available in the article and its supplementary information file. Source data are provided with this paper. Source data in this paper have been deposited in Figshare with a DOI (https://doi.org/10.6084/m9.figshare.22736900). Source data are provided with this paper.

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

## Acknowledgements

We thank Dr. Xueling Huang, Dr. Hua Zhao, and Dr. Fengping Yuan (National Key Laboratory of Crop Improvement for Stress Tolerance and Production, Northwest A&F University, Yangling, China) for technique assistance. This research was funded by a grant from the National Natural Science Foundation of China (32222008 to C.W, 32100215 to PP.H).

## Author contributions

This research was conceived by C.W. Z. Y, SY. G, Q. L, YJ, F, PP. H, and CF. J performed the experiments. Z.Y analyzed the data. C.W and Z.Y wrote the manuscript with comments from all authors.

## Competing interests

The authors declare no competing interests.
