## [Peer Review File · Nature Communications]

The CBL1/9-CIPK1 calcium sensor negatively regulates drought stress by phosphorylating the PYLs ABA receptorREVIEWER COMMENTS

Reviewer #1 (Remarks to the Author):

The authors present some data to show that CIPK1 may phosphorylate some of the ABA receptors to block their function in ABA signaling. This is an interesting idea and some interesting data are shown. However, the data overall are not sufficient to support the author's conclusions, as described below.

Specific comments:

1. Line 111-112: This assay does not measure drought tolerance as the authors have not done anything to measure soil water potential or water content and nothing to ensure that all genotypes are exposed to the same severity of soil drying. The results are consistent with their other assays in that genotypes that do not close their stomata as efficiently and transpire more deplete soil water faster and thus look worse by the end of the period of uncontrolled soil drying. But this is not drought tolerance (which can only be assayed if all genotypes are exposed to same severity of soil drying), it is just a change water use and should be referred to as drought avoidance rather than drought tolerance.
2. Line 144-145: The BiFC assay as it is done here cannot really be quantified. This assay alone is insufficient to say that there is interaction with some PYLs and not with others (and then use this to exclude some of the PYLs from all further analyses).
3. Line 152-153: Empty vector is not appropriate as a negative control for split-LUC or BiFC assays. What about the rest of the PYLs in this assay?
4. Figure 2E: Why is there no GFP band in the input blot for the GFP negative control? This is not a negative control unless you actually show that GFP protein could be detected in the input.
5. Line 177-179: The authors say that the phosphorylation assay was consistent with the interaction assay, yet several of the PYLs (PYR1, PYL2, PYL3) that interacted with CIPK1 did not show any phosphorylation. How do the authors explain this?
6. Line 204-206: Yeast two hybrid assay is not sufficient to say that phosphomimic PYL4 could not interact with ABI1. GST-pull down using PYL4 phosphonull and mimic forms are required at a minimum to establish this point (more quantitative assay of binding affinity such as SPR would be preferred).
7. Line 228: this needs to be written more precisely to make clear that this is of phosphomimic PYL at one specific phosphorylation site, with other sites presumably having no effect.
8. Figure 5A: This one very unclear IP result is not sufficient to show that ABA inhibits interaction of PYL4 and CIPK. More clear result, and preferably the use of quantitative methods which can actually show difference in binding affinity, need to be presented. Also, what happens with CIPK1 interaction with other PYLs? Is it also affected by ABA?
9. Fig. 5E: The model shown here is interesting, but is also premature and needs to be supported by more and better data.
10. Line 292-293: The manuscript does not contain any measurements of growth during drought stress. Thus, there is no basis for the authors to make any statement about the balance between growth and drought response.
11. Line 319: This was only shown for one PYL. I think the authors would have to show that the same effect of phosphomimic/phosphonull version of the protein for at least one additional PYL before they can try to generalize their conclusions to cover all the PYLs.
12. Line 347: Here again, no measurements of growth during drought stress are presented (as a start, such assays would need to have the wild type and mutant actually exposed to the same severity of stress, which was not done in the soil drying experiments shown). Thus there is no basis to talk about CBL1/9-CIPK optimizing growth versus stress response. All they have shown is an effect on transpiration, likely caused by change in stomatal behavior,

and nothing else.

Reviewer #2 (Remarks to the Author):

The authors investigated interaction of CBL1/9-activated CIPK1 with ABA receptors and phosphorylation of PYLs to clarify the role of CIPKs in ABA responses. The calcium sensor phosphorylated PYLs by directly interacting with ABA receptors (PYR1/PYL1-6), the phosphorylation of PYLs inhibited their activities, and Ser129 in PYL4 was essential for PYLs to bind ABA and inhibit PP2C. The authors claimed that the negative regulation is important for preventing stress signaling by ABA, that ABA-independent PYLs in the absence of stress, and that under stress conditions, ABA inhibits PYLs phosphorylation by CIPK1, activating downstream ABA signaling and ensuring plant survival. This is an interesting topic. However, there are several concerns in this study.

English editing is required.

The authors did stomatal assay. The stomatal apertures were about 6 μm , which are larger than the stomatal apertures (2 μm to 3 μm) previously reported using Arabidopsis. I guess that the authors do not measure stomatal pores. If the data is correct, the authors should show the picture of stomata with scale bar.

Furthermore, the solution for stomatal closure assay contains 50 mM CaCl_2 , which is so high that stomatal closure is induced. The stomatal data is questionable and doubtful.

Regarding their statistical analysis, did the authors compared the averages? The authors mention just usage of ANOVA in the text. The authors should also compare the averages.

Figure 5: the authors used CIPK but not BCL/CIPK complex, that is, the CIPK is not activated by BCL. Hence, the data does not support their conclusion that BCL/CIPK phosphorylates PYLs under normal condition.

Reviewer #3 (Remarks to the Author):

This work provides further evidence to link ABA and calcium stress signaling pathways in plants, and because of this, this study is of broad interest and importance. A number of prior reports have demonstrated a clear link between CBL-CIPK complexes and components of ABA signaling and response, including drought response. Furthermore, there is also some previous evidence that CBP-CIPK proteins might regulate stress pathways via direct interaction with PYL type ABA receptors – e.g. Xu et al. 2020 (Environmental and Experimental Botany vol. 172, 103999) demonstrated that a CIPK from *Vitis amurensis* can bind to a PYL protein, and may therefore control drought responses via this pathway. However, such regulation has not been validated elsewhere, and there are still aspects of a CBL-CIPK PYR/PYL protein regulation process that have not yet been characterized. This manuscript goes some way to addressing some of the open questions, although not all gaps

are filled.

The authors should make reference to all relevant prior studies, including the aforementioned study of Xu et al. (2020).

The first experiment presented in Figure 1 is useful validation for use of the *Arabidopsis thaliana* Col-0 background, even though this is largely confirmatory data to that shown elsewhere, such as in *Arabidopsis thaliana* cv. WS (D'Angelo et al., 2006). However, I have concerns about the plant images that are presented. How is it possible for the WT (Col-0) plants to display such distinct morphological drought responses when exposed to apparently identical conditions ("Two-week-old seedlings growing in the greenhouse were subjected to drought stress by withholding water for 20 d. Then, they were rewatered for 3 d and imaged" – Fig. 1C showing relatively good drought tolerance by the WT while Fig. 1F shows clear drought sensitivity by the WT plants). Then in Fig. 5D we see very little difference between drought stressed WT and *cipk1-3* mutant, which contradicts what we see in Fig. 1B, although the imposition of drought seems to be a few days longer in Fig. 1B). If this indeed indicates substantial natural variation between drought responses then more representative images as well as quantitative data (e.g., plant biomass measurements) must be shown. In fact, I would prefer to also see quantitative data alongside the photographic images for all plant growth phenotypic data in the manuscript. Note that Fig. 1D is referred to as Fig. 1C in the main text.

In my view the data supporting the interaction between CIPK1 and specific PYL proteins as well as the phosphorylation by CIPK1 is convincing, however, there are a few critical aspects that should be addressed. Firstly, can the authors confirm where in the cell the CIPK1 interaction is taking place? It has been previously demonstrated that CBL1-CIPK1 and CBL9-CIPK1 can localize to the plasma membrane, so is this also the site of CIPK1-PYL interaction? Secondly, is there a CBL and Ca²⁺ dependency to the CIPK1-mediated PYL phosphorylation, particularly since in many cases the CBL partner regulates kinase activity of the CIPK following Ca²⁺ binding? In my view, this is critical to demonstrate that it is indeed a CBL1/9-CIPK1 calcium sensor that is regulating PYL proteins by phosphorylation in response to Ca²⁺ feedback, rather than lone CIPK1 binding. As it stands, the statement that there is a "Ca²⁺-CBL1/9-CIPK1-PYLs signaling pathway, which is probably involved in negative feedback of Ca²⁺ on ABA signaling" is premature without further experiments.

Response to the reviewer's comments:

For clarity, our responses are written in blue color.

REVIEWER COMMENTS

Reviewer #1 (Remarks to the Author):

The authors present some data to show that CIPK1 may phosphorylate some of the ABA receptors to block their function in ABA signaling. This is an interesting idea and some interesting data are shown. However, the data overall are not sufficient to support the author's conclusions, as described below.

Response: We are very grateful for recognizing our work's value and helping us improve the manuscript.

Specific comments:

1. Line 111-112: This assay does not measure drought tolerance as the authors have not done anything to measure soil water potential or water content and nothing to ensure that all genotypes are exposed to the same severity of soil drying. The results are consistent with their other assays in that genotypes that do not close their stomates as efficiently and transpire more deplete soil water faster and thus look worse by the end of the period of uncontrolled soil drying. But this is not drought tolerance (which can only be assayed if all genotypes are exposed to same severity of soil drying), it is just a change water use and should be referred to as drought avoidance rather than drought tolerance.

Response: Thank you very much for your sensible comments. In light of your valuable suggestions, we conducted additional experiments to ensure consistency and accuracy of our findings. Specifically, we repeated all drought experiments and measured the relative water content of each small pot to verify that the plants were exposed to the same drought conditions (Supplemental Fig.1). Additionally, we ensured that all small pots contained the same weight of soil during the drought assays, and moved the pots regularly to maintain consistency during the drought treatment. Furthermore, we analyzed the fresh weight of detached leaves and the leaf surface temperature of different plants, which supported our initial findings (Fig. 1). These newly added results provide further evidence to reinforce our previous conclusions. Once again, thank you

very much for your valuable input.

Supplemental Fig.1 Measurement of soil water content during drought treatments.

(A-E) Pot weight for drought assays were measured and plotted as a percentage of the original weight.

Fig.1. Phenotype of *cbl1/9*, *cipk1* mutants, and *CIPK1* overexpression lines. (A) GUS Histochemical staining in *ProCIPK1: GUS* lines. Scale bar, 20 μ m. (B) *cipk1* mutants showed enhanced drought tolerance compared to the wild type. Seven-day-old seedlings were transferred to small pots. Each pot contains 9 seedlings. After 7 d of growth under sufficient water, seedlings were subjected to drought stress by withholding water for 20 d, then rewatered for 3 days. Pictures were taken at each stage of treatment. (C) Water loss measurement in detached rosette leaves of 4-week-old plants. The experiment was repeated three times with independent treatments. Data are means \pm SD of 3 replicates. ** $P < 0.01$, *** $P < 0.001$, Student's t-test. (D) ABA-mediated stomatal closure in *cipk1* mutants. Leaves from 3-week-old plants were immersed in an MES-KOH buffer under light conditions for 2.5 h to open the stomata completely. Then, they were transferred to an MES-KOH buffer with 10 μ M ABA for 2 h and imaged. (E) Stomatal aperture (μm) for Control and 10 μ M ABA treatments. (F) Leaf temperature (°C) for Col-0, *cipk1-2*, and *cipk1-3*. (G) Leaf temperature (°C) for Col-0, *cipk1-2*, and *cipk1-3*. (H) Relative water loss (%) over 3 hours for Col-0, *CIPK1 CA#1*, and *CIPK1 CA#2*. (I) Stomatal closure for Col-0, *CIPK1 CA#1*, and *CIPK1 CA#2*. (J) Stomatal aperture (μm) for Col-0, *CIPK1 CA#1*, and *CIPK1 CA#2*. (K) Leaf temperature (°C) for Col-0, *CIPK1 CA#1*, and *CIPK1 CA#2*. (L) Photos of *cbl1/9* and *cbl1/9/cipk1* mutants under different treatments. (M) Photos and heatmap of leaf temperature (°C) for Col-0, *cipk1-2*, and *cbl1/9/cipk1*. (N) Leaf temperature (°C) for Col-0, *cipk1-2*, and *cbl1/9/cipk1*. (O) Leaf temperature (°C) for Col-0, *CIPK1 CA#1*, and *CIPK1 CA#2*.

(E) Statistical analysis of ABA-induced stomatal closure in wild type, *cipk1-2*, and *cipk1-3* plants. The box and whiskers plots (Tukey method) represent minimum and maximum values. Statistical analysis was performed by two-way analysis of variance (ANOVA) followed by Dunnett's multiple comparison test. The asterisk indicates significance compared to wild type: ** P<0.01. The experiment was repeated three times as different biological replicates with a minimum of 100 stomatal pores.

(F) Leaf temperature of *cipk1-2* and *cipk1-3* mutants determined by infrared thermography. Four-week-old plants grown in soil (soil moisture was 75%-85%) were photographed with an infrared camera. The temperature of 9 leaves was measured in triplicate. Statistical analysis was performed by one-way analysis of variance (ANOVA) followed by Dunnett's multiple comparison test. Asterisk indicates significance compared to wild type (** P < 0.01).

(G) *CIPK1CA* drought sensitivity compared to the wild type. Two-week-old seedlings grown in the greenhouse were subjected to drought stress by withholding water for 20 d. Then, rewatered for 3 d and imaged. The data shown represent three independent experiments.

(H) Water loss of *CIPK1CA* plant leaves. The experiment was repeated three times with independent treatments. Data are means \pm SD of 3 replicates. Asterisks indicate significance compared to the wild type according to Student's t-test (***) P < 0.001).

(I) ABA-mediated stomatal closure in *CIPK1CA* mutants. Leaves from 3-week-old plants were immersed in an MES-KOH buffer under light for 2.5 h to open the stomata completely. Then, transferred to an MES-KOH buffer supplemented with 10 μ M ABA for 2 h and imaged.

(J) Statistical analysis of ABA-induced stomatal closure in wild type, *CIPK1CA#1*, and *CIPK1CA#2* plants. The box and whiskers plots (Tukey method) represent minimum and maximum values. Different asterisks indicate statistically different means by Dunnett's multiple comparisons tests. The asterisk indicates significance compared to wild-type: ** P<0.01, Two-way ANOVA.

(K) Leaf temperature of *CIPK1CA#1* and *CIPK1CA#2* plants determined by infrared thermography. Four-week-old plants grown in soil (moisture 65%-75%). Leaf temperature was measured in nine leaves in triplicate. Statistical

analysis was performed by one-way analysis of variance (ANOVA) followed by Dunnett's multiple comparison test. Asterisks indicate significance compared to wild type (** P < 0.01).

(L) *cb1/9* and *cb1/9/cipk1* drought tolerance phenotype. Two-week-old seedlings grown in the greenhouse were subjected to drought stress by withholding water for 22 d, then, rewatered for 3 days. Pictures were taken at each stage.

(M) The leaf temperature of *cb1/9* and *cb1/9/cipk1* plants was determined using infrared thermography. Four-week-old plants grown in soil (moisture was 65%-75%) were photographed with an infrared camera. Leaf temperature was measured in nine leaves in triplicate. Statistical analysis was performed by one-way analysis of variance (ANOVA) followed by Dunnett's multiple comparison test. Asterisk indicates significance compared to wild type: ** P<0.01.

2. Line 144-145: The BiFC assay as it is done here cannot really be quantified. This assay alone is insufficient to say that there is interaction with some PYLs and not with others (and then use this to exclude some of the PYLs from all further analyses).

Response: Thank you very much for your valuable comments. In response to your comments, we have removed the quantitative results for BiFC in Fig 2. To further validate the interactions between PYLs and CIPK1, we conducted split-LUC assays and obtained similar results that support our hypothesis. Specifically, our data demonstrated that CIPK1 only interacted with PYR1/PYL1-PYL6 (Fig. 2C and Supplemental Fig. 2B). Additionally, we have used GST-PYL9 as a negative control in our pull-down assay, which further confirmed that CIPK1 only interacts with PYR1/PYL1-PYL6 (Fig. 2D).

(C) CIPK1 interacts with PYR1/PYL1-PYL6. CIPK1-nLUC and PYLs-cLUC were co-infiltrated into tobacco leaves. Signals were detected by CCD after 3 d. GUS-nLUC and GUS-cLUC were used as negative controls.

(D) CIPK1 interacts with PYR1/PYL1-PYL6. Recombinant GST or GST-PYLs were incubated with glutathione Sepharose beads for 2 h. Then, beads were incubated with CIPK1-His for another 2 h and washed six times to eliminate nonspecific binding. Anti-GST and Anti-His were used for protein detection.

(B) CIPK1 does not interact with PYL7-13 in LUC complementation imaging (LCI) assay.

3. Line 152-153: Empty vector is not appropriate as a negative control for split-LUC or BiFC assays. What about the rest of the PYLs in this assay?

Response: Thank you very much for your comments. We agree that the empty vector is inappropriate as a negative control. By using BiFC and Split-LUC assays, we found that CIPK1 does not interact with PYL7-PYL13. We have added these results in Fig.2A and Supplemental Fig. 2B.

CIPK1 does not interact with PYL7-13 in BiFC and LUC complementation imaging (LCI) assay.

4. Figure 2E: Why is there no GFP band in the input blot for the GFP negative control? This is not a negative control unless you actually show that GFP protein could be detected in the input.

Response: Thank you very much for your comments. The level of GFP protein mixed with various other proteins was significantly more abundant in the input as compared to the CIPK1-GFP fusion protein, which is why we did not show the GFP protein bands in the initial version. In the revised version, we co-expressed CIPK1-FLAG with PYL1-GFP or PYL4-GFP in *Arabidopsis*, and the

outcomes demonstrated that CIPK1-FLAG could successfully interact with both PYL1-GFP and PYL4-GFP. We have added these results in Fig.2E.

(E) Co-IP assay showing the interaction of CIPK1 with PYL1 and PYL4.

5. Line 177-179: The authors say that the phosphorylation assay was consistent with the interaction assay, yet several of the PYLs (PYR1, PYL2, PYL3) that interacted with CIPK1 did not show any phosphorylation. How do the authors explain this?

Response: Thank you very much for your comments. As indicated in Fig. 3A, phosphorylation bands for PYL2 and PYL3 were observed, albeit faintly, and have been identified using a red arrow. Moreover, our phosphorylation assay on GST-tagged PYR1 and PYL2 demonstrated that CIPK1 can trigger their phosphorylation. We have incorporated these findings into Fig. 3A and Supplemental Fig. 3A.

(A) CIPK1 phosphorylation of PYLs *in vitro*. In the assay, 1.5 μ g CIPK1-His and \sim 2 μ g PYLs-His were incubated for 30 min at 30°C in a kinase buffer with 2 μ Ci [γ -³²P] ATP. The proteins were separated by 12% SDS-PAGE. Autoradiography (upper panel) and Coomassie staining (lower panel) show the

phosphorylation and protein loading of the PYLs, respectively.

(A) CIPK1 phosphorylation of PYR1 and PYL2 *in vitro*. In the assay, 1 μ g CIPK1-His and 1.5 μ g GST-PYLs were incubated for 30 min at 30°C in a kinase buffer supplemented with 2 μ Ci [γ - 32 P] ATP. The proteins were separated by 12% SDS-PAGE.

6. Line 204-206: Yeast two hybrid assay is not sufficient to say that phosphomimic PYL4 could not interact with ABI1. GST-pull down using PYL4 phosphonull and mimic forms are required at a minimum to establish this point (more quantitative assay of binding affinity such as SPR would be preferred).

Response: Thank you very much for your comments. Here, we utilized pull-down assays to validate our findings and observed that phosphorylated PYL1 and PYL4 were unable to bind to ABI1. We have added these results in Fig. 3F and Supplemental Fig. 3D.

(F) PYL4^{S129D} did not interact with ABI1 in pull-down assay.

(D) PYL1^{T133D} did not interact with ABI1 in pull-down assay.

7. Line 228: this needs to be written more precisely to make clear that this of phosphomimic PYL at one specific phosphorylation site, with other sites presumably having no effect.

Response: Thank you very much for your comments. We have revised the original sentence to “Based on the *in vitro* findings and the results obtained from transgenic plants, it can be inferred that phosphomimetic mutated PYLs located at the conserved site corresponding to PYL4 Ser129 are not active receptors.” to improve the accuracy of the description.

8. Figure 5A: This one very unclear IP result is not sufficient to show that ABA inhibits interaction of PYL4 and CIPK. More clear result, and preferably the use of quantitative methods which can actually show difference in binding affinity, need to be presented. Also, what happens with CIPK1 interaction with other PYLs? Is it also affected by ABA?

Response: Thank you very much for your comments. We also investigated the impact of ABA on the interaction between CIPK1 and PYL1/PYL4 through pull-down and MST assays. Our findings indicated that the presence of ABA led to

a decrease in the binding affinity of CIPK1 towards PYL1 and PYL4. We have added these results to Fig. 5 and Supplemental Fig. 7.

(A) ABA treatment inhibits the interactions between CIPK1 and PYL4.

(B) ABA impairs the affinity of PYL4 with CIPK1 in MST assays. 100 μM ABA was added to the reaction system and incubated on ice for 1 h.

Supplemental Fig. 7. Inhibition of the interaction between CIPK1 and PYL1 by ABA.

(A) Increasing concentrations of ABA inhibits the interaction between CIPK1 and PYL1 in pull-down assays.

(B) ABA impairs the affinity of PYL1 by CIPK1 in MST assays. 100 μM ABA was added to the reaction system and incubated on ice for 1 h.

9. Fig. 5E: The model shown here is interesting, but is also premature and needs to be supported by more and better data.

Response: Thank you very much for your comments. Based on your comments, we added the relevant experimental results, which validate our previous findings and strengthen our conclusions. Furthermore, we have updated the model presented in Figure 6.

Fig. 6. A proposed model illustrating how the phosphorylation of ABA receptors by CIPK1 acts as a negative regulator of drought stress responses in plants. Under normal conditions, CIPK1 acts to suppress the activation of ABA responses by phosphorylating PYL4 at Ser129, which inactivates the ABA receptor and prevents it from inhibiting the phosphatase activities of ABI1/PP2Cs. This results in a lack of downstream ABA signaling. However, under drought stress conditions, ABA inhibits CBL1/9-CIPK1-mediated phosphorylation of PYR1/PYL1-6, and active ABA receptors interact with ABI1/PP2Cs to inhibit their phosphatase activity, leading to the activation of downstream ABA signaling. This response enables plants to survive under drought stress. Positive regulation is indicated by arrows with solid lines, while strong negative regulation is represented by a solid line with a horizontal bar.

10. Line 292-293: The manuscript does not contain any measurements of growth during drought stress. Thus, there is no basis for the authors to make any statement about the balance between growth and drought response.

Response: Thank you very much for your comments. To avoid exaggeration, we have removed certain statements. Instead, we have conducted experiments to support our findings. Specifically, we measured the fresh weight of detached leaves from *PYL4* transgenic plants, photographed the wilting status of detached rosette leaves from various ecotypes at a specific time, and measured the leaf surface temperature of these plants. All of these experiments revealed that phosphorylation resulted in ABA tolerance and drought sensitivity of *PYL4*. These results have been included in Fig. 4.

(D) Phenotype of *PYL4/12458*, *PYL4^{S129A}/12458*, and *PYL4^{S129D}/12458* under drought conditions.

(E) Phenotype of detached leaves of *PYL4* transgenic plants.

(F) water loss measure in detached leaves of *PYL4* transgenic plants. The experiment was repeated three times with independent treatments. Data are means \pm SD of 3 replicates. *** $P < 0.001$, Student's t-test.

(G) Leaf temperature of *PYL4* transgenic plants. Photographs were taken using an infrared thermal imager at relative soil water content between 70% and 80%. Leaf temperature was measured on nine leaves in triplicates. Statistical analysis was performed by one-way analysis of variance (ANOVA) followed by Dunnett's multiple comparison test. Asterisks indicate significance compared to

wild type (** P < 0.01).

11. Line 319: This was only shown for one PYL. I think the authors would have to show that the same effect of phosphomimic/phosphonull version of the protein for at least one additional PYL before they can try to generalize their conclusions to cover all the PYLs.

Response: Thank you very much for your comments. To address your concerns, we conducted an *in vitro* phosphatase activity assay to explore how phosphorylated mutant forms of PYL1 affect ABI1. In addition, via pull-down assays, we found no evidence of phosphorylated PYL1 binding to ABI1. These findings suggest that the phosphorylation of PYLs by CIPK1 is a common phenomenon. We have added these results in Supplemental Fig. 3.

(C) PYL1^{T133D} did not inhibit the phosphatase activity of ABI1.

(D) Phosphomimetic PYL1 cannot interact with ABI1.

12. Line 347: Here again, no measurements of growth during drought stress are presented (as a start, such assays would need to have the wild type and mutant actually exposed to the same severity of stress, which was not done in the soil drying experiments shown). Thus there is no basis to talk about CBL1/9-CIPK optimizing growth versus stress response. All they have shown is an effect on transpiration, likely caused by change in stomatal behavior, and nothing else.

Response: Thank you very much for your comments. To ensure the plants were

exposed to similar drought conditions, we measured the relative soil water content during the experiment. Additionally, we examined the fresh weight of detached leaves and the leaf temperature of the plants, and our findings were consistent. In addition, to avoid exaggeration, we have removed certain statements regarding optimizing growth versus stress response.

(D) Phenotype of *PYL4/12458*, *PYL4^{S129A}/12458*, and *PYL4^{S129D}/12458* under drought conditions.

(E) Phenotype of detached leaves of *PYL4* transgenic plants.

(F) water loss measure in detached leaves of *PYL4* transgenic plants. The experiment was repeated three times with independent treatments. Data are

means \pm SD of 3 replicates. Asterisks indicate significance compared to wild type. *** $P < 0.001$, Student's t-test.

(G) Leaf temperature of *PYL4* transgenic plants. Photographs were taken using an infrared thermal imager at relative soil water content between 70% and 80%. Leaf temperature was measured on nine leaves in triplicates. Statistical analysis was performed by one-way analysis of variance (ANOVA) followed by Dunnett's multiple comparison test. Asterisks indicate significance compared to wild type (** $P < 0.01$).

Reviewer #2 (Remarks to the Author):

The authors investigated interaction of CBL1/9-activated CIPK1 with ABA receptors and phosphorylation of PYLs to clarify the role of CIPKs in ABA responses. The calcium sensor phosphorylated PYLs by directly interacting with ABA receptors (PYR1/PYL1-6), the phosphorylation of PYLs inhibited their activities, and Ser129 in PYL4 was essential for PYLs to bind ABA and inhibit PP2C. The authors claimed that the negative regulation is important for preventing stress signaling by ABA, that ABA-independent PYLs in the absence of stress, and that under stress conditions, ABA inhibits PYLs phosphorylation by CIPK1, activating downstream ABA signaling and ensuring plant survival. This is an interesting topic. However, there are several concerns in this study.

Response: We are very grateful for recognizing our work and helping us improve our manuscripts.

English editing is required.

Response: Thank you very much for your valuable comments. Our language has been professionally polished.

The authors did stomatal assay. The stomatal apertures were about 6 μm , which are larger than the stomatal apertures (2 μm to 3 μm) previously reported using Arabidopsis. I guess that the authors do not measure stomatal pores. If the data is correct, the authors should show the picture of stomata with scale bar.

Response: Thank you very much for your valuable comments. We apologize for any confusion caused. Our measurements indicate that all plants had stomata apertures of approximately 5-6 μm in the control state. This could be attributed to the use of intense light, promoting complete stomata opening. Moreover, we have included representative pictures to validate our findings. We have added these results in Fig.1. Once again, thank you very much for your valuable input.

(D) ABA-mediated stomatal closure in *cipk1* mutants. Leaves from 3-week-old plants were immersed in an MES-KOH buffer under light conditions for 2.5 h to open the stomata completely. Then, they were transferred to an MES-KOH buffer with 10 μ M ABA for 2 h and imaged.

(I) ABA-mediated stomatal closure in *CIPK1CA* mutants. Leaves from 3-week-old plants were immersed in an MES-KOH buffer under light for 2.5 h to open the stomata completely. Then, transferred to an MES-KOH buffer supplemented with 10 μ M ABA for 2 h and imaged.

Furthermore, the solution for stomatal closure assay contains 50 mM CaCl₂, which is so high that stomatal closure is induced. The stomatal data is questionable and doubtful.

Response: Thank you very much for your comments. We apologize for the typo in which we mistakenly wrote 50 mM instead of 50 μ M. We have rectified the error in the materials and methods section.

Regarding their statistical analysis, did the authors compare the averages? The authors mention just usage of ANOVA in the text. The authors should also compare the averages.

Response: Thank you very much for your comments. ANOVA was used to calculate statistical differences. Asterisks indicate means that were statistically different by Dunnett's multiple comparisons tests. All the information has been modified in the figure legends.

Figure 5: the authors used CIPK but not BCL/CIPK complex, that is, the CIPK is not activated by BCL. Hence, the data does not support their conclusion that BCL/CIPK phosphorylates PYLs under normal condition.

Response: Thank you very much for your comments. We used *in vivo* phosphorylation experiments and found that in *cb1/9* mutant, *PYL4* and *PYL4^{S129A}* showed similar phosphorylation levels, while in *PYL4* plants, *PYL4* phosphorylation showed a gradual decrease, consistent with the results of

semi-in vivo phosphorylation. These results demonstrate that the CBL/CIPK complex regulates the phosphorylation of PYLs. We have added these results in Fig.5.

(F) CBL1/9-CIPK1 mediates the phosphorylation of PYL4 Ser129 *in vivo*. 10-d-old seedlings were treated with 60 μ M ABA for the indicated times. The phosphorylation levels PYL4-FLAG and PYL4-FLAG loading were detected by Anti-P-Ser antibodies and Anti-FLAG antibody respectively.

Reviewer #3 (Remarks to the Author):

This work provides further evidence to link ABA and calcium stress signaling pathways in plants, and because of this, this study is of broad interest and importance. A number of prior reports have demonstrated a clear link between CBL-CIPK complexes and components of ABA signaling and response, including drought response. Furthermore, there is also some previous evidence that CBP-CIPK proteins might regulate stress pathways via direct interaction with PYL type ABA receptors – e.g. Xu et al. 2020 (Environmental and Experimental Botany vol. 172, 103999) demonstrated that a CIPK from *Vitis amurensis* can bind to a PYL protein, and may therefore control drought responses via this pathway. However, such regulation has not been validated elsewhere, and there are still aspects of a CBL-CIPK PYR/PYL protein regulation process that have not yet been characterized. This manuscript goes some way to addressing some of the open questions, although not all gaps are filled.

Response: We are very grateful for the recognition of our work and for helping us improve our manuscript.

The authors should make reference to all relevant prior studies, including the aforementioned study of Xu et al. (2020).

Response: Thank you very much for your comments. We have cited this article.

The first experiment presented in Figure 1 is useful validation for use of the *Arabidopsis thaliana* Col-0 background, even though this is largely confirmatory data to that shown elsewhere, such as in *Arabidopsis thaliana* cv. WS (D'Angelo et al., 2006). However, I have concerns about the plant images that are presented. How is it possible for the WT (Col-0) plants to display such distinct morphological drought responses when exposed to apparently identical conditions (“Two-week-old seedlings growing in the greenhouse were subjected to drought stress by withholding water for 20 d. Then, they were rewatered for 3 d and imaged” – Fig. 1C showing relatively good drought tolerance by the WT while Fig. 1F shows clear drought sensitivity by the WT plants). Then in Fig. 5D we see very little difference between drought stressed WT and *cipk1-3* mutant, which contradicts what we see in Fig. 1B, although the imposition of drought seems to be a few days longer in Fig. 1B). If this indeed indicates substantial natural variation between drought responses then more representative images as well as quantitative data (e.g., plant biomass measurements) must be shown. In fact, I would prefer to also see quantitative data alongside the photographic images for all plant growth phenotypic data in the manuscript. Note that Fig. 1D is referred to as Fig. 1C in the main text.

Response: Thank you very much for your valuable comments. Our earlier experiments were designed to highlight the differences, which resulted in varying phenotypes exhibited by the wild-type plants in different trials. However, following your constructive input, we have re-run the drought experiments and meticulously monitored the soil water content across different pot sizes to ensure uniformity in the phenotypes of all wild-type plants. Moreover, we have included measurements of detached leaf fresh weight and leaf surface temperature in multiple trials, which unequivocally validate the precision of our experimental output. To reflect these updated findings, we have incorporated the new data into Fig. 1, Supplemental Fig. 1, Fig. 4, and Fig. 5. Once again, thank you very much for your valuable input.

Fig.1. Phenotype of *cbl1/9*, *cipk1* mutants, and *CIPK1* overexpression lines. (A) GUS Histochemical staining in *ProCIPK1: GUS* lines. Scale bar, 20 μ m. (B) *cipk1* mutants showed enhanced drought tolerance compared to the wild type. Seven-day-old seedlings were transferred to small pots. Each pot contains 9 seedlings. After 7 d of growth under sufficient water, seedlings were subjected to drought stress by withholding water for 20 d, then rewatered for 3 days. Pictures were taken at each stage of treatment. (C) Water loss measurement in detached rosette leaves of 4-week-old plants. The experiment was repeated three times with independent treatments. Data are means \pm SD of 3 replicates. ** $P < 0.01$, *** $P < 0.001$, Student's t-test. (D) ABA-mediated stomatal closure in *cipk1* mutants. Leaves from 3-week-old plants were immersed in an MES-KOH buffer under light conditions for 2.5 h to open the stomata completely. Then, they were transferred to an MES-KOH buffer with 10 μ M ABA for 2 h and imaged. (E) Stomatal aperture (μ m) in Col-0, *cipk1-2*, and *cipk1-3* under control and 10 μ M ABA. (F) Leaf temperature ($^{\circ}$ C) in Col-0, *cipk1-2*, and *cipk1-3* before, during, and after drought. (G) Leaf temperature ($^{\circ}$ C) in Col-0, *cipk1-2*, and *cipk1-3*. (H) Relative water loss (%) in Col-0, *CIPK1 CA#1*, and *CIPK1 CA#2* over 3 hours. (I) Stomatal aperture (μ m) in Col-0, *CIPK1 CA#1*, and *CIPK1 CA#2* under control and 10 μ M ABA. (J) Stomatal aperture (μ m) in Col-0, *CIPK1 CA#1*, and *CIPK1 CA#2*. (K) Leaf temperature ($^{\circ}$ C) in Col-0, *CIPK1 CA#1*, and *CIPK1 CA#2*. (L) Photos of seedlings before, during, and after drought for Col-0, *cipk1-2*, *cbl1/9*, and *cbl1/9/cipk1*. (M) Leaf temperature ($^{\circ}$ C) in Col-0, *cipk1-2*, *cbl1/9*, and *cbl1/9/cipk1*.

(E) Statistical analysis of ABA-induced stomatal closure in wild-type, *cipk1-2*, and *cipk1-3* plants. The box and whiskers plots (Tukey method) represent minimum and maximum values. Statistical analysis was performed by two-way analysis of variance (ANOVA) followed by Dunnett's multiple comparison test. The asterisk indicates significance compared to wild type: ** P<0.01. The experiment was repeated three times as different biological replicates with a minimum of 100 stomatal pores.

(F) Leaf temperature of *cipk1-2* and *cipk1-3* mutants determined by infrared thermography. Four-week-old plants grown in soil (soil moisture was 75%-85%) were photographed with an infrared camera. The temperature of 9 leaves was measured in triplicate. Statistical analysis was performed by one-way analysis of variance (ANOVA) followed by Dunnett's multiple comparison test. Asterisk indicates significance compared to wild type (** P < 0.01).

(G) *CIPK1CA* drought sensitivity compared to the wild type. Two-week-old seedlings grown in the greenhouse were subjected to drought stress by withholding water for 20 d. Then, rewatered for 3 d and imaged. The data shown represent three independent experiments.

(H) Water loss of *CIPK1CA* plant leaves. The experiment was repeated three times with independent treatments. Data are means \pm SD of 3 replicates. Asterisks indicate significance compared to the wild type according to Student's t-test (***) P < 0.001).

(I) ABA-mediated stomatal closure in *CIPK1CA* mutants. Leaves from 3-week-old plants were immersed in an MES-KOH buffer under light for 2.5 h to open the stomata completely. Then, transferred to an MES-KOH buffer supplemented with 10 μ M ABA for 2 h and imaged.

(J) Statistical analysis of ABA-induced stomatal closure in wild type, *CIPK1CA#1*, and *CIPK1CA#2* plants. The box and whiskers plots (Tukey method) represent minimum and maximum values. Different asterisks indicate statistically different means by Dunnett's multiple comparisons tests. The asterisk indicates significance compared to wild-type: ** P<0.01, Two-way ANOVA.

(K) Leaf temperature of *CIPK1CA#1* and *CIPK1CA#2* plants determined by infrared thermography. Four-week-old plants grown in soil (moisture 65%-75%). Leaf temperature was measured in nine leaves in triplicate. Statistical

analysis was performed by one-way analysis of variance (ANOVA) followed by Dunnett's multiple comparison test. Asterisks indicate significance compared to wild type (** P < 0.01).

(L) *cb11/9* and *cb11/9/cipk1* drought tolerance phenotype. Two-week-old seedlings grown in the greenhouse were subjected to drought stress by withholding water for 22 d, then, rewatered for 3 days. Pictures were taken at each stage.

(M) The leaf temperature of *cb11/9* and *cb11/9/cipk1* plants was determined using infrared thermography. Four-week-old plants grown in soil (moisture was 65%-75%) were photographed with an infrared camera. Leaf temperature was measured in nine leaves in triplicate. Statistical analysis was performed by one-way analysis of variance (ANOVA) followed by Dunnett's multiple comparison test. Asterisk indicates significance compared to wild type: ** P<0.01.

Supplemental Fig.1 Measurement of soil water content during drought treatments.

(A-E) Pot weight for drought assays were measured and plotted as a percentage of the original weight.

(D) Phenotype of *PYL4/12458*, *PYL4^{S129A}/12458*, and *PYL4^{S129D}/12458* under drought conditions.

(E) Phenotype of detached leaves of *PYL4* transgenic plants.

(F) Water loss measure in detached leaves of *PYL4* transgenic plants. The experiment was repeated three times with independent treatments. Data are means \pm SD of 3 replicates. Asterisks indicate significance compared to wild type. *** $P < 0.001$, Student's t-test.

(G) Leaf temperature of *PYL4* transgenic plants. Photographs were taken using an infrared thermal imager at relative soil water content between 70% and 80%. Leaf temperature was measured on nine leaves in triplicates. Statistical analysis was performed by one-way analysis of variance (ANOVA) followed by Dunnett's multiple comparison test. Asterisks indicate significance compared to wild type (** $P < 0.01$).

(G) The *12458/cipk1* mutants showed similar drought sensitivity. Seven-day-old seedlings were transferred to small pots, with 9 seedlings in each pot. After 7 d of growth under sufficient water, seedlings were subjected to drought stress by withholding water for 18 d.

(H) *12458* and *12458/cipk1* plants leaf temperature imaging. Photographs were taken using an infrared thermographer at relative soil moisture contents between 75% and 85%. Data were statistical for each replicate of 9 leaves repeated three times. Statistical analysis was performed by one-way analysis of variance (ANOVA) followed by Dunnett's multiple comparison test. Asterisks indicate significance compared to wild type (** $P < 0.01$).

In my view the data supporting the interaction between CIPK1 and specific PYL proteins as well as the phosphorylation by CIPK1 is convincing, however, there are a few critical aspects that should be addressed. Firstly, can the authors confirm where in the cell the CIPK interaction is taking place? It has been previously demonstrated that CBL1-CIPK1 and CBL9-CIPK1 can localize to the plasma membrane, so is this also the site of CIPK1-PYL interaction?

Response: Thank you very much for your comments. In light of your comments, we have undertaken additional experiments and incorporated them into our study. Specifically, we performed protoplast BiFC assays and discovered that CIPK1 and PYLs interact in the cytoplasm, nucleus, and cell membrane. However, the assay outcome might not entirely correspond to the location of the actual interaction. Hence, we included CBL1-FLAG in the interaction system and established that the interaction between CIPK1 and PYLs happens exclusively at the cell membrane. This signifies that the CIPK1-PYLs interaction occurs in plants at the cytoplasmic membrane. We have added these results in Fig. 2.

(B) CBL1 anchors the interaction of CIPK1 and PYL4 to the plasma membrane. Tobacco leaf protoplasts injected with different combinations were observed under confocal microscopy. Scale bar, 20 μ m.

Secondly, is there a CBL and Ca²⁺ dependency to the CIPK1-mediated PYL phosphorylation, particularly since in many cases the CBL partner regulates kinase activity of the CIPK following Ca²⁺ binding? In my view, this is critical to demonstrate that it is indeed a CBL1/9-CIPK1 calcium sensor that is regulating

PYL proteins by phosphorylation in response to Ca²⁺ feedback, rather than lone CIPK1 binding. As it stands, the statement that there is a “Ca²⁺-CBL1/9-CIPK1-PYLS signaling pathway, which is probably involved in negative feedback of Ca²⁺ on ABA signaling” is premature without further experiments.

Response: Thank you very much for your comments. In order to ascertain the potential involvement of Ca²⁺ in CIPK1-mediated phosphorylation of PYLs, we carried out *in vitro* phosphorylation experiments. Our findings indicated that the presence of EGTA in the reaction adversely affected the phosphorylation of PYLs by CIPK1. Additionally, the introduction of Ca²⁺ into the reaction revealed that high concentrations of Ca²⁺ had an impact on the phosphorylation of PYL4 by CIPK1, hinting that the phosphorylation of PYL4 likely occurs at lower cytosolic Ca²⁺ levels. Our *in vivo* phosphorylation assays further confirmed CBL1/9-CIPK1-mediated phosphorylation of PYL4. Specifically, the results demonstrated that the level of phosphorylation of PYL4 was significantly diminished upon ABA treatment. We have added these results in Fig.5.

(E) Effects of Ca²⁺ on the Phosphorylation of PYL4 by CIPK1.

(F) CBL1/9-CIPK1 mediates the phosphorylation of PYL4 Ser129 *in vivo*. 10-d-old seedlings were treated with 60 μM ABA for the indicated times. The phosphorylation levels PYL4-FLAG and PYL4-FLAG loading were detected by Anti-P-Ser antibodies and Anti-FLAG antibody respectively.

REVIEWER COMMENTS

Reviewer #1 (Remarks to the Author):

The reviewers have added new data which have improved the manuscript. However, there are still some points to address. One major issue is that the authors continue to speak of drought tolerance when they have no evidence for it. The soil water content data added in supplemental figure 1 shows obvious differences in water content on the low end of the soil drying curve. Because the relationship between soil water content and water potential is not linear, these differences in soil water content which may seem small actually indicate a substantial difference in water content. And it is also simply not plausible that the growth of the plants in small pots, there are obvious differences in water usage and leaf temperature and then somehow this does not lead to differences in soil water content? It is simply not believable. And they also show that one of the CIPKs is just expressed in stomata. The authors should remove mention of drought tolerance and simply describe the differences in leaf water loss and leaf temperature that they do appropriately document. The manuscript overall will be stronger and more clear with this more appropriate interpretation of the soil drying data.

The MST analysis is a good addition but is not conclusive. The relatively small change in K_d seems not consistent with the large changes in other assays. In particular, it is not clear that K_d difference between plus and minus ABA is statistically significant, especially for PYL1 given the wide variation in the data. The authors need to improve their data (additional replicates perhaps) and add appropriate controls (such as PYL1/4 interaction with ABI1 and how the K_d of that interaction is modified by ABA) to show that their MST assays are working. In that case they could also validate whether phosphorylated PYL really has reduced K_d for interaction with phosphatase. Also, the MST assays are not described in sufficient detail in the methods.

The authors also need to more clearly discuss and acknowledge what is already known about PYL ABA binding. Many researchers believe that PYLs and Clade A PP2Cs act as co-receptors. ABA binding requires both the PYL and the phosphatase. Thus, it would be unexpected that ABA affects the binding of PYL to other protein. Thus, the authors need to provide more conclusive evidence to be convincing and also must discuss how their data can be reconciled with previous analyses of PYL ABA binding.

The writing overall needs to be improved and clarified. The newly added parts in particular contain unclear sentences and errors in wording. A few examples (but not all) are pointed out below.

Other points:

1. There is no label for "G" in Figure 1.
2. Line 175-177: if this experiment does not indicate the *in vivo* localization of the interaction, then why do the BiFC experiment in this way?
3. Line 200, 202: presumably a FLAG tag was used instead of FALG?
4. Line 206 first word should be "of" or "by"?
5. Images in all the figures need to include scale bars. Fig 4E adds nothing and could be deleted.
6. Line 319: unclear what is meant by "necessitates" in this context.
7. Fig 5F is not conclusive and the authors have cropped the gel images so closely that it is not clear whether their assay is really specifically detecting phosphorylated PYL4 and whether the bands they are showing us are at the expected molecular weight. Hard to believe that they really detected no phosphorylation in the *cb1/9* plants given how much phosphorylation they seem to have detected in wild type background (ie: there are other *cb1s* which we may expect to have some effect, even if more minor than CBL1/9).

8. Line 421: this statement not justified since there are no quantitative measurements of plant growth in this manuscript.
9. Blots in Fig 2D have been overadjusted for brightness/contrast.

Reviewer #2 (Remarks to the Author):

The authors revised the manuscript but not sufficiently.

The authors added the representative pictures showing stomata to Fig. 1. However, there is no explanation of the scale bar. Moreover, the authors did not make it clear where the authors measured as a stomatal aperture. Hence, the stomatal aperture data is still questionable. The authors should add a scale bar and should indicate where they measured as a stomatal aperture in the picture.

Reviewer #3 (Remarks to the Author):

The authors have made a substantial number of revisions to their manuscript with new data that now much more convincingly supports their conclusions. In particular, there is now clearer evidence of the role of this CBL-CIPK module in regulating drought resilience in a calcium dependent manner and through the action of ABA signaling. I feel that the changes made to the manuscript have suitably addressed all of my original comments and concerns.

Not all of the methods details for the newly added experiments seem to have been included. These need to be provided. For example, I cannot see the methods details for the IR thermography or for the water loss measurements of detached leaves.

Please be consistent in providing details of the numbers of biological and/or technical replicates used for all of the experiments in the figure legends, as well as the number of repetitions for representative images, as these details are not always provided.

What was the plasma membrane marker used for the protoplast BiFC assay? Please provide the details.

For the new text inserted in lines 163-165 please revise the grammar – e.g. remove “may”.

In lines 200 and 202, “FALG” should be FLAG

Response to the reviewer's comments:

For clarity, our responses are written in blue color.

REVIEWER COMMENTS

Reviewer #1 (Remarks to the Author):

The reviewers have added new data which have improved the manuscript. However, there are still some points to address. One major issue is that the authors continue to speak of drought tolerance when they have no evidence for it.

Response: Thank you very much for recognizing our revised manuscript and providing valuable comments to improve it. As per your suggestion, we have modified the terminology used in our manuscript from "drought tolerance" to "drought resilience" or "water retention capacity."

The soil water content data added in supplemental figure 1 shows obvious differences in water content on the low end of the soil drying curve. Because the relationship between soil water content and water potential is not linear, these differences in soil water content which may seem small actually indicate a substantial difference in water content. And it is also simply not plausible that the growth of the plants in small pots, there are obvious differences in water usage and leaf temperature and then somehow this does not lead to differences in soil water content? It is simply not believable.

Response: Thank you very much for your sensible comments. As you pointed out, the variation in relative soil water content at later stages of drought in Supplementary Figure 1D and E can be attributed to the increased vulnerability of the ABA receptor quintuple mutant (*pyr1pyl2458*) to drought stress. As a result, we can observe changes in soil water content in the smaller pots corresponding to the *pyr1pyl2458* mutant, *PYL4^{S129D}* transgenic plant, and *12458/cipk1* mutant. However, compared to the ABA receptor quintuple mutant, the soil moisture content in other small potted plants remained largely unchanged. This is because the drought resilience of these mutants is more similar, with only minor differences, unlike the ABA receptor quintuple mutant. And they also show that one of the CIPKs is just expressed in stomata.

Response: Thank you very much for your comments. We apologize for any

confusion caused. In the manuscript, we stated that CIPK1 is highly expressed in stomata, but did not intend to imply that it is exclusively expressed only in stomata. It has been previously reported that CIPK1 shares a similar expression pattern with CBL1/9 (D'Angelo et al., 2006), with the latter being highly expressed in stomata (Cheong et al., 2007). However, the expression of CIPK1 in stomata has not been documented. To address this, we conducted an analysis of the stomatal expression pattern of CIPK1 using a native promoter-driven *GUS* reporter gene of *CIPK1* and validated our findings by observing its high expression in stomata.

The authors should remove mention of drought tolerance and simply describe the differences in leaf water loss and leaf temperature that they do appropriately document. The manuscript overall will be stronger and more clear with this more appropriate interpretation of the soil drying data.

Response: We are very grateful for your valuable comments and help us improve the manuscript. We have removed mention of “drought tolerance”. In this version, we revised the “drought tolerance” to “water retention capacity” or “drought resilience”.

The MST analysis is a good addition but is not conclusive. The relatively small change in K_d seems not consistent with the large changes in other assays. In particular, it is not clear that K_d difference between plus and minus ABA is statistically significant, especially for PYL1 given the wide variation in the data. The authors need to improve their data (additional replicates perhaps) and add appropriate controls (such as PYL1/4 interaction with ABI1 and how the K_d of that interaction is modified by ABA) to show that their MST assays are working. In that case they could also validate whether phosphorylated PYL really has reduced K_d for interaction with phosphatase. Also, the MST assays are not described in sufficient detail in the methods.

Response: We are very grateful for your valuable comments and help us improve the manuscript. Based on your feedback, we utilized PYL1/4 in combination with ABI1 as a control measure to confirm the suitability of the system. Furthermore, besides PBST (PBS+0.05%) buffer which was mentioned in the previous manuscript, we also explored the use of other buffers in our experiments (Ye et al., 2017; Wang et al., 2018). However, none of these buffers yielded satisfactory results. In combination with the unremarkable change in K_D

values of the MST results in our previous manuscript, we made the decision to exclude the relevant data from the manuscript to prevent any potential errors in our MST results.

Our *in vitro* Pull-down and Co-IP findings provided conclusive evidence that ABA inhibits the interactions between CIPK1 and PYLs. Furthermore, our *in vitro* phosphorylation results demonstrated that the presence of ABA directly reduced the phosphorylation intensity of PYL4, while leaving the phosphorylation intensity of CIPK1 unaltered. Taken together, these results suggest that ABA is capable of inhibiting the interaction between CIPK1 and PYLs, ultimately promoting ABA signaling during drought conditions. The Co-IP results have been added to Fig. 5 and Supplemental Fig. 7.

(A) ABA treatment inhibits the interaction between CIPK1 and PYL4 in pull-down assays.

(B) ABA inhibits the interaction between CIPK1 and PYL4 in Co-IP assays. For ABA treatment, the extracted protein containing 100 μM ABA was incubated overnight with Anti-FLAG antibodies.

(C) ABA inhibits the phosphorylation of PYL4 by CIPK1 *in vitro*. In this assay, 1.5 μg CIPK1-His, 2 μg PYLs-His, and indicated concentrations of ABA in kinase buffer were incubated with 2 μCi [γ - ^{32}P] ATP for 30 min at 30°C.

(B) ABA inhibits the interaction between CIPK1 and PYL1 in Co-IP assays.

The authors also need to more clearly discuss and acknowledge what is already known about PYL ABA binding. Many researchers believe that PYLs and Clade A PP2Cs act as co-receptors. ABA binding requires both the PYL and the phosphatase. Thus, it would be unexpected that ABA affects the binding of PYL to other protein. Thus, the authors need to provide more conclusive evidence to be convincing and also must discuss how their data can be reconciled with previous analyses of PYL ABA binding.

Response: Thank you very much for this valuable comment. Structural studies highlight the conserved gate-latch-lock mechanism of ABA perception and signal transduction (Melcher et al., 2009; Miyazono et al., 2009; Nishimura et al., 2009; Santiago et al., 2009; Yin et al., 2009). When the ABA receptor binds ABA, the conformational change allows the ligand to entry the gate onto the latch, allowing the receptor to dock with PP2Cs. A tryptophan conserved in the PP2C active site is in turn inserted between the gate and the latch, further locking the receptor-PP2C complex (Melcher et al., 2009). The capacity of the ABA receptors PYLs protein to bind ABA is nearly 100-fold higher when clade A PP2C proteins are present (Zhu, 2016). Therefore, PP2Cs can be considered as a co-receptor for ABA, which have an important role in the stabilization of ABA signaling and ABA-PYLs-PP2Cs complexes in plants (Ma et al., 2009).

In this manuscript, it was observed that phosphorylated PYLs exhibited impaired binding to PP2C. Consequently, this could negatively impact the binding of ABA and its resultant signaling. The structural changes elicited by ABA binding in the ABA receptor could potentially affect the binding affinity of CIPK1 to PYLs. In this case, ABA acts to inhibit the interaction between CIPK1 and the ABA receptor to promote efficient ABA signaling. Furthermore, prior

research has demonstrated that ABA is capable of inhibiting both the interaction and phosphorylation of PYLs by EL1-like casein kinases to prevent the degradation of the ABA receptors (Chen et al., 2018). This part of the discussion has been added to the Discussion section, as seen in lines 435-448.

The writing overall needs to be improved and clarified. The newly added parts in particular contain unclear sentences and errors in wording. A few examples (but not all) are pointed out below.

Other points:

1. There is no label for “G” in Figure 1.

Response: Thank you very much for your careful review. We have added the label “G” in Figure 1.

2. Line 175-177: if this experiment does not indicate the *in vivo* localization of the interaction, then why do the BiFC experiment in this way?

Response: Thank you very much for your valuable comments. We apologize for the ambiguity caused by this sentence, and we have removed it from the manuscript.

3. Line 200, 202: presumably a FLAG tag was used instead of FALG?

Response: Thank you for your careful review. We misspelled "FLAG" as "FALG", which we have revised in lines 199 and 201 of the manuscript.

4. Line 206 first word should be “of” or “by”?

Response: Thank you very much for pointing this out. we have made a revision in line 205 of the manuscript.

5. Images in all the figures need to include scale bars. Fig 4E adds nothing and could be deleted.

Response: Thank you very much for your comments. According to your comments, we have added scale bars to the images. In addition, Fig. 4E has been removed from the manuscript.

6. Line 319: unclear what is meant by “necessitates” in this context.

Response: Thank you very much for your comments. We apologize for the confusion caused by the “necessitates”. We misspelled "necessitates" as “necessitates”. What we mean here is " requires ". We have revised it to "required" in line 319 of the manuscript. The following are the modifications in the MS: To further elucidate if CIPK1-mediated phosphorylation of PYLs requires CBL1/9, an *in vivo* phosphorylation assay was conducted.

7. Fig 5F is not conclusive and the authors have cropped the gel images so closely that it is not clear whether their assay is really specifically detecting phosphorylated PYL4 and whether the bands they are showing us are at the expected molecular weight. Hard to believe that they really detected no phosphorylation in the *cb1/9* plants given how much phosphorylation they seem to have detected in wild type background (ie: there are other cbls which we may expected to have some effect, even if more minor than CBL1/9).

Response: Thank you very much for your comments. We apologize for any confusion that may have been caused by the gel images. Following your suggestion, we performed several replicates and found that we did observe faint phosphorylation bands in *PYL4/cb1/9* and *PYL4^{S129A}* plants when increasing the exposure time and the amount of loading samples. The result has been added in Fig. 5F.

(F) CBL1/9-CIPK1 mediates the phosphorylation of PYL4 Ser129 *in vivo*. 10-day-old seedlings were treated with 60 μ M ABA for the indicated times. The phosphorylation signal of PYL4 was detected by anti-P-Ser antibodies, and the proteins were quantified through western blot and anti-FLAG antibodies.

8. Line 421: this statement not justified since there are no quantitative measurements of plant growth in this manuscript.

Response: Thank you very much for your comments. We have removed the description of the growth in the manuscript.

9. Blots in Fig 2D have been overadjusted for brightness/contrast.

Response: Thank you very much for your careful review. Based on your comment, we have replaced the image of Fig 2D.

(D) Pull-down assay showing CIPK1 interaction with PYR1/PYL1-PYL6. Recombinant GST or GST-PYs were incubated with Glutathione Sepharose beads for 2 h. Then, beads were incubated with CIPK1-His for another 2 h and washed six times to eliminate non-specific binding. Anti-GST and Anti-His were used for protein detection.

Reviewer #2 (Remarks to the Author):

The authors revised the manuscript but not sufficiently.

The authors added the representative pictures showing stomata to Fig. 1. However, there is no explanation of the scale bar. Moreover, the authors did not make it clear where the authors measured as a stomatal aperture. Hence, the stomatal aperture data is still questionable. The authors should add a scale bar and should indicate where they measured as a stomatal aperture in the picture.

Response: Thank you for your careful review and your valuable comment. We apologize for neglecting to indicate the length of the scale bar and using the wrong method to measure stomatal apertures in Figure 1. Furthermore, based on your feedback and consultation with peers, we have corrected the stomatal aperture measurements and highlighted our measured stomatal widths on representative stomata with dashed lines.

(D) ABA-mediated stomatal closure in *cipk1* mutants. Leaves from 3-week-old plants were immersed in an MES-KOH buffer under light conditions for 2.5 h to open the stomata completely. Then, they were transferred to an MES-KOH buffer with 10 μM ABA for 2 h and imaged. Stomatal width measurements are marked in the figure using red dashed lines. Scale bar, 10 μm.

(E) Statistical analysis of ABA-induced stomatal closure in wild-type, *cipk1-2*, and *cipk1-3* plants. The box and whiskers plots (Tukey method) represent minimum and maximum values. Statistical analysis was performed by two-way analysis of variance (ANOVA) followed by Dunnett's multiple comparison test. The asterisk indicates significance compared to the wild type: ** P<0.01. The experiment was repeated three times as different biological replicates with a minimum of 100 stomatal pores.

(I) ABA-mediated stomatal closure in *CIPK1CA* mutants. Leaves from 3-week-old plants were immersed in an MES-KOH buffer under light for 2.5 h to open the stomata completely. Then, transferred to an MES-KOH buffer supplemented with 10 μM ABA for 2 h and imaged. Stomatal width measurements are marked in the figure using red dashed lines. Scale bar, 10 μm.

(J) Statistical analysis of ABA-induced stomatal closure in wild type, *CIPK1CA#1*, and *CIPK1CA#2* plants. The box and whiskers plots (Tukey method) represent minimum and maximum values. Different asterisks indicate statistically different means by Dunnett's multiple comparisons tests. The asterisk indicates significance compared to wild-type: ** P<0.01, Two-way

ANOVA. The experiment was repeated three times as different biological replicates with a minimum of 100 stomatal pores.

Reviewer #3 (Remarks to the Author):

The authors have made a substantial number of revisions to their manuscript with new data that now much more convincingly supports their conclusions. In particular, there is now clearer evidence of the role of this CBL-CIPK module in regulating drought resilience in a calcium dependent manner and through the action of ABA signaling. I feel that the changes made to the manuscript have suitably addressed all of my original comments and concerns.

Response: We are very grateful for recognizing our work's value and help us improve the manuscript.

Not all of the methods details for the newly added experiments seem to have been included. These need to be provided. For example, I cannot see the methods details for the IR thermography or for the water loss measurements of detached leaves.

Response: Thank you very much for your careful review. We have added the details of the methods for the IR thermography or the water loss measurements of detached leaves into MATERIALS AND METHODS.

Please be consistent in providing details of the numbers of biological and/or technical replicates used for all of the experiments in the figure legends, as well as the number of repetitions for representative images, as these details are not always provided.

Response: Thank you very much for your careful review and your valuable comment. All the details of the numbers of biological and/or technical replicates used for all of the experiments have been added to the figure legends.

What was the plasma membrane marker used for the protoplast BiFC assay? Please provide the details.

Response: Thank you very much for your careful review. We have incorporated the detail for the plasma membrane marker in the figure legend, as described in lines 911-913. Specifically, we utilized the CBL1n-OFP construct, which involves the fusion of the first 12 N-terminal amino acids of CBL1 with GFP, serving as the plasma-membrane marker (PM Marker).

For the new text inserted in lines 163-165 please revise the grammar – e.g. remove “may”.

Response: Thank you very much for your suggestions. We have removed the “may” in line 163 of the manuscript.

In lines 200 and 202, “FALG” should be FLAG

Response: Thank you very much for your careful review. We misspelled "FLAG" as "FALG", which we have revised in lines 199 and 201 of the manuscript.

References

- Chen, H.-H., Qu, L., Xu, Z.-H., Zhu, J.-K., and Xue, H.-W.** (2018). EL1-like casein kinases suppress ABA signaling and responses by phosphorylating and destabilizing the ABA receptors PYR/PYLs in *Arabidopsis*. *Molecular Plant* **11**, 706-719.
- Cheong, Y.H., Pandey, G.K., Grant, J.J., Batistic, O., Li, L., Kim, B.G., Lee, S.C., Kudla, J., and Luan, S.** (2007). Two calcineurin B-like calcium sensors, interacting with protein kinase CIPK23, regulate leaf transpiration and root potassium uptake in *Arabidopsis*. *The Plant Journal* **52**, 223-239.
- D'Angelo, C., Weinl, S., Batistic, O., Pandey, G.K., Cheong, Y.H., Schültke, S., Albrecht, V., Ehlert, B., Schulz, B., and Harter, K.** (2006). Alternative complex formation of the Ca²⁺-regulated protein kinase CIPK1 controls abscisic acid-dependent and independent stress responses in *Arabidopsis*. *The Plant Journal* **48**, 857-872.
- Ma, Y., Szostkiewicz, I., Korte, A., Moes, D., Yang, Y., Christmann, A., and Grill, E.** (2009). Regulators of PP2C phosphatase activity function as abscisic acid sensors. *Science* **324**, 1064-1068.
- Melcher, K., Ng, L.-M., Zhou, X.E., Soon, F.-F., Xu, Y., Suino-Powell, K.M., Park, S.-Y., Weiner, J.J., Fujii, H., and Chinnusamy, V.** (2009). A gate-latch-lock mechanism for hormone signalling by abscisic acid receptors. *Nature* **462**, 602-608.
- Miyazono, K.-i., Miyakawa, T., Sawano, Y., Kubota, K., Kang, H.-J., Asano, A., Miyauchi, Y., Takahashi, M., Zhi, Y., and Fujita, Y.** (2009). Structural basis of abscisic acid signalling. *Nature* **462**, 609-614.
- Nishimura, N., Hitomi, K., Arvai, A.S., Rambo, R.P., Hitomi, C., Cutler, S.R., Schroeder, J.I., and Getzoff, E.D.** (2009). Structural mechanism of abscisic acid binding and signaling by dimeric PYR1. *Science* **326**, 1373-1379.
- Santiago, J., Dupeux, F., Round, A., Antoni, R., Park, S.-Y., Jamin, M., Cutler, S.R., Rodriguez, P.L., and Márquez, J.A.** (2009). The abscisic acid receptor PYR1 in complex with abscisic acid. *Nature* **462**, 665-668.
- Wang, P., Zhao, Y., Li, Z., Hsu, C.-C., Liu, X., Fu, L., Hou, Y.-J., Du, Y., Xie, S., and Zhang, C.** (2018). Reciprocal regulation of the TOR kinase and

ABA receptor balances plant growth and stress response. *Molecular cell* **69**, 100-112. e106.

Ye, Y., Zhou, L., Liu, X., Liu, H., Li, D., Cao, M., Chen, H., Xu, L., Zhu, J.-k., and Zhao, Y. (2017). A novel chemical inhibitor of ABA signaling targets all ABA receptors. *Plant Physiology* **173**, 2356-2369.

Yin, P., Fan, H., Hao, Q., Yuan, X., Wu, D., Pang, Y., Yan, C., Li, W., Wang, J., and Yan, N. (2009). Structural insights into the mechanism of abscisic acid signaling by PYL proteins. *Nature structural & molecular biology* **16**, 1230-1236.

Zhu, J.-K. (2016). Abiotic stress signaling and responses in plants. *Cell* **167**, 313-324.

REVIEWERS' COMMENTS

Reviewer #1 (Remarks to the Author):

The authors have responded adequately to most comments. The weak part of the manuscript is still the authors assertion that ABA directly affects CIPK-PYL interaction and phosphorylation. The main data supporting this is now Fig 5A-D but none of these experiments included quantitation and none give any indication of whether the result shown is reproducible. Showing graphs with quantitation of band intensities from 3 or more replicate experiments along with appropriate statistical analysis would help better support the authors conclusions.

Also, something is wrong with the sentence on line 255-256.

Reviewer #2 (Remarks to the Author):

The authors revised the manuscript.

The authors changed their stomatal data from about 6 um to about 3 um without any reason. Why was the data changed? The authors did not mention the difference in Materials and Methods.

Reviewer #3 (Remarks to the Author):

All of my previous comments have been suitably addressed and I have no further comments for this manuscript.

Point to point response to reviewers

Response to the Editor and the Reviewer's comments:

For clarity, our responses are written in blue color.

REVIEWER COMMENTS

Reviewer #1 (Remarks to the Author)

The authors have responded adequately to most comments. The weak part of the manuscript is still the authors assertion that ABA directly affects CIPK-PYL interaction and phosphorylation. The main data supporting this is now Fig 5A-D but none of these experiments included quantitation and none give any indication of whether the result shown is reproducible. Showing graphs with

quantitation of band intensities from 3 or more replicate experiments along with appropriate statistical analysis would help better support the authors conclusions.

Response: Thank you very much for your careful review and your valuable comment. Each of the four experiments was conducted a minimum of two to three times, consistently yielding comparable outcomes. To enhance the reliability of the source data, we have included an additional three repetitions and conducted statistical analysis on the Co-IP relative intensities. The resulting data has been then incorporated into Fig. 5B, C. Taken together, these four results suggest that ABA is capable of inhibiting the interaction between CIPK1 and PYLs, ultimately promoting ABA signaling during drought conditions.

(B) ABA inhibits the interaction between CIPK1 and PYL4 in Co-IP assays. For ABA treatment, the extracted protein containing 100 μ M ABA was incubated overnight with Anti-FLAG antibodies. Three repetitions of the experiment obtained similar results.

(C) The relative density of PYL4 (IP/Input) reveals in (B) an interaction between CIPK1 and PYL4 *in vivo* that was reduced by ABA. Data were measured using Image J and represented as IP/Input. Data are shown as means \pm SD. Asterisks indicate significance compared to the IP without ABA according to Student's t-test (** P < 0.01).

Also, something is wrong with the sentence on line 255-256.

Response: Thank you very much for your careful review. We have modified the sentence to "In contrast, transgenic lines expressing *PYL4*^{S129D} exhibited the ABA-insensitive phenotype of 12458 mutant plants." in line 255-256.

Reviewer #2 (Remarks to the Author):

The authors revised the manuscript.

The authors changed their stomatal data from about 6 μm to about 3 μm without any reason. Why was the data changed? The authors did not mention the difference in Materials and Methods.

Response: Thank you very much for your careful review and your valuable comment. We deeply apologize for the error made in the earlier manuscript, where we mistakenly identified the length indicated by the dashed line in the left image as the stomatal width. However, after consulting with our peers and considering the feedback from the reviewer, we have rectified this issue. We conducted new measurements to accurately determine the stomatal widths, and we have replaced the representative images with the corrected versions in order to ensure the reliability and accuracy of our data (shown in the right image).

The data for each ecotype was collected from three repetitions, with a total of no less than 100 stomata measured. The Source data file has been updated to reflect these measurements in the last version, and we have also included the specific methods used to measure stomatal aperture in the Materials and Methods section, which is described as follows “The minimum pore space between the two guard cells that make up the stomata was the stomata width. Finally, the stomatal apertures were measured using the Image J software. The data for each ecotype was collected from three repetitions, with a total of no less than 100 stomata measured”.

The dashed line, positioned between the two black lines, represents the width measurement of the stomata in various manuscripts. The left panel showcases the initial findings, whereas the right panel exhibits the amended depiction of stomatal measurements.

Reviewer #3 (Remarks to the Author)

All of my previous comments have been suitably addressed and I have no further comments for this manuscript.

Response: We are very grateful for recognizing our work's value and help us improve the manuscript.